

# North Atlantic Oscillation polarity during the past 3 ka derived from lacustrine sediments of large lowland lake Schweriner See, NE-Germany

Marie-Luise Adolph[1*], Sambor Czerwinski[1,2], Mirko Dreßler[1], Paul Strobel[3], Marcel Bliedtner[3], Sebastian Lorenz[1], Maxime Debret[4], Torsten Haberzettl[1]

[1] Department of Physical Geography, Institute for Geography and Geology, University of Greifswald, Germany
[2] Climate Change Ecology Research Unit, Faculty of Geographical and Geological Sciences, Adam Mickiewicz University in Poznań, Poland
[3] Department of Physical Geography, Institute for Geography, Friedrich-Schiller-University Jena, Germany
[4] UMR 6143 M2C Laboratoire Morphodynamique Continentale et Côtière, Department of Geoscience, Université de Rouen Normandie, France

*Correspondence to*: Marie-Luise Adolph (marie-luise.adolph@uni-greifswald.de)

**Abstract.** Based on a multi-dating and multi-proxy approach, we reconstruct Late Holocene environmental changes from sediments of Schweriner See, a large lowland lake in NE-Germany spanning the past $3070^{+170}/_{-210}$ cal BP. We infer large-scale atmospheric variations using a combination of in-lake productivity indicators using traditional and high-resolution techniques (e.g. $LOI_{550}$, TOC, inc/coh), diatom assemblages, which are sensitive to ice-cover duration, as well as compound-specific hydrogen isotopes ($\delta^2H_{C25}$) reflecting variability in the moisture source region distinguishing the southern and northern North Atlantic and/or Arctic region and/or the degree of evaporative lake water enrichment. Our study shows that before 1850 CE, in-lake productivity at Schweriner See was mainly influenced by winter temperature variability, which modulates ice-cover duration and growing-season length. Low productivity co-occurs with the occurrence of the diatom species *Stephanocostis chantaicus*, which blooms below the ice cover, indicating temporal prolonged ice cover duration. Simultaneously, changes to a moisture source region in the northern North Atlantic and/or Arctic regions and/or low evaporative lake water enrichment are inferred from $\delta^2H_{C25}$. In contrast, high productivity is linked to the disappearance of *S. chantaicus* and moisture originating from the southern North Atlantic and/or high evaporative lake water enrichment. These distinct changes are driven by variations between positive and negative NAO polarity during the past $3070^{+170}/_{-210}$ cal BP. Besides these long-term shifts in atmospheric conditions, short-term variations can be inferred from titanium concentrations, which mainly reflect paleo-shoreline distance likely linked to precipitation variability and, after the 12th century, to anthropogenic impacts. Since 1850 CE, productivity has been driven by nutrient availability.



# 1 Introduction

In recent decades, droughts have increased in frequency and severity in Central Europe (Spinoni et al., 2018) with severe socio-economic and ecological consequences. Future climate scenarios for Western and Central Europe predict increasing temperatures, more frequent, longer and/or more intense heat waves, as well as warm spells and an increase in dryness with short-term droughts (IPCC, 2021) affecting the hydrological cycle and, therefore, all aquatic (eco)systems. Lowering lake and groundwater levels in Central Europe, such as in Northeast Germany (Germer et al., 2010), has already affected some areas in Central Europe. However, to understand the drivers, magnitude and direction of climatic and environmental changes and to assess future developments, longer time series than those provided by monitoring efforts are needed (IPCC, 2021). Existing studies from North Germany point to considerable environmental variability during the Holocene (e.g. Dietze et al., 2016; Theuerkauf et al., 2022; Kaiser et al., 2012) and a spatial climatic gradient with increasing continentality from west to east, which is influenced by internal dynamics of the climate system such as the North Atlantic Oscillation (NAO) governing precipitation and surface air temperature (Hurrell, 2003). However, an in-depth understanding of the Holocene hydroclimatic variability of North Germany is still limited because the majority of studies have been carried out in more continental areas (e.g. Dietze et al., 2016; Lampe et al., 2009; Lorenz, 2007; Theuerkauf et al., 2022) and studies from the transition from maritime to continental conditions are rare (e.g. Lorenz, 2007). Moreover, many studies have been carried out on small lacustrine systems (e.g. Dreßler et al., 2011), in which anthropogenic impact often overprints natural climate variations (Haberzettl et al., 2019). These biases culminate in sometimes contradicting results, e.g. reconstructed lake-level curves, which have been used as key tools for hydroclimatic reconstructions (Kaiser et al., 2012). Apart from that, several studies from that area stress that not all observed lake-level variations are induced by climatic variations but rather by (anthropogenic) landcover changes influencing evapotranspiration and, consequently, groundwater recharge (e.g. Theuerkauf et al., 2022; Dietze et al., 2016).

In a wider regional context, hydroclimatic variability in the North Atlantic region during the Late Holocene has been related to changes in ocean circulations (e.g. Trouet et al., 2012; Bond et al., 2001), solar cycles (e.g. Martin-Puertas et al., 2012; Mellström et al., 2015) and atmospheric modes such as the NAO. The NAO is one of the leading atmospheric circulation systems influencing weather and climate conditions in the Northern Hemisphere (Bliedtner et al., 2023; Hurrell and Deser, 2009). It refers to changes in the atmospheric mass balance, i.e. the air pressure difference between the subpolar low (Iceland) and subtropic high (Azores) sea-level pressure (SLP) systems over the North Atlantic. These, in turn, influence surface air temperature, precipitation, and wind and storminess, including wind direction and storm tracks (Hu et al., 2022). NAO varies between two modes depending on the barometric difference between the pressure systems. A positive NAO (NAO+) is associated with a stronger gradient between the pressure systems causing zonal circulation and increased intensity of cyclones resulting, e.g. in stronger Westerlies (Hurrell and Deser, 2009). In contrast, a negative NAO (NAO-) has a weaker gradient causing a meridional circulation with weaker Westerlies, which results in more frequent atmospheric blocking and allows colder air from the Arctic regions to flow towards Europe (Hurrell and Deser, 2009). The NAO is more active in the cold



season (October–April), with larger amplitudes and a strong influence on winter temperature and precipitation (Hurrell et al., 2003). Generally, a positive NAO is associated with mild and moist conditions (maritime), while a negative NAO is associated with cold and dry (continental) conditions in Central Europe (Hurrell and Deser, 2009). Recently, it has been suggested that winter conditions in the North Atlantic region can be linked to the combined effects of the NAO and the second mode of variability, i.e. the Eastern Atlantic pattern (EA) (e.g. Comas-Bru and McDermott, 2014; Mellado-Cano et al., 2019). The EA pattern is defined as sea-level pressure monopole between Iceland and Ireland (e.g. Comas-Bru and McDermott, 2014; Moore et al., 2013) modulating the location and intensities of the Icelandic Low and Azores High (e.g. Moore et al., 2011) consequently modulating e.g. the position of the North Atlantic storm tracks and jet stream (Woollings et al., 2010; Seierstad et al., 2007; Moore and Renfrew, 2012).

Coastal areas surrounding the Baltic Sea were identified as ideal for collecting proxy information about large-scale North Atlantic atmospheric patterns (e.g. the NAO) because temporal non-stationarities are neglectable there (Comas-Bru et al., 2016). To investigate the impacts of these large-scale atmospheric circulation patterns on North German lowlands, we identified Schweriner See (See = lake), a large hard-water lake located approximately 20 km south of the Baltic Sea and close to the boundary between more maritime to more continental climate, as a suitable archive that is not affected by the biases in previous studies in small lacustrine systems. As a rather large lake, Schweriner See is less susceptible to anthropogenic biases. Schweriner See has a relatively small catchment compared to its size (Wöbbecke et al., 2003), making it sensitive to hydrological changes.

## 2 Study Area

Schweriner See (53°43.256'N 11°27.544'E, 37.8 m a.s.l.) is a hard-water lake located in the North German lowlands in the westernmost part of the Mecklenburg Lake District and approx. 20 km south of the Baltic Sea (Fig. 1). The lake has a surface area of 61.54 km$^2$, extends over 24.8 km in the N-S direction and is up to 6 km wide in the E-W direction. Nowadays, Schweriner See has two similar-sized basins separated by an (in parts) artificial dam (Paulsdamm, Fig. 1A), which was built to connect the western and eastern shorelines in 1848 CE (Kasten and Rost, 2005). This was made possible by a lake-level decline initiated by the broadening of the main outflow of the Stör waterway, which exposed the so-called Ramper Moor (Fig. 1). As a consequence, the previously periodically flooded Ramper Moor peninsula emerged as a calcareous mire (Umweltministerium Mecklenburg-Vorpommern, 2003). The area of the dam is characterised by strong carbonate-rich groundwater inflow that results in an increased carbonate precipitation (Fig. 1C, Adolph et al., 2023). Before these construction activities, both lake basins were openly connected (Wiebeking, 1786), but today, they are only linked by a small passage. Both lake basins are characterised by complex morphometry with several deep areas, steep slopes, channel structures and extended shallow water areas (Fig. 1). The sediment core investigated in this study was taken in the deepest spot (52 m water depth) of the northern basin, the so-called Schweriner Außensee (SAS), which is characterised by a large shallow water area (< 5 m water depth) in the eastern littoral area influenced by wave- and wind-induced dynamics (Fig. 1C, Adolph et al., 2023). This





area divides Schweriner Außensee into two subbasins in the south and north, whose depositional processes are mainly influenced by carbonate precipitation and productivity (Fig. 1, Adolph et al., 2023).

The overall catchment area is 414 km², but Schweriner Außensee comprises only 85 km². Overall, the catchment is mainly composed of farmland (47.5 %), water surfaces (20.9 %), forests (12.8 %), and populated areas (10.9 %) (Wöbbecke et al., 2003). The lake basin is mainly fed by groundwater (~70 %, pers. communication M. Lückstädt, Staatliches Amt für

Landwirtschaft und Umwelt Westmecklenburg) and precipitation. Surrounding smaller lakes and tributaries are of minor importance for Schweriner Innensee (southern basin, Fig. 1A, Nixdorf et al., 2004) and Schweriner Außensee has only few small inflowing streams (Fig. 1B). Nowadays Schweriner See has two outflows passing ice marginal positions (IMP) of the Weichselian glaciation at the southern and northern end between which Schweriner See is located (Krienke and Obst, 2011). At the southern end of Schweriner See, the river Stör drains through a valley formed by glacial meltwaters that broke through

the southern IMP. The artificial Wallenstein trench was built in the 16$^{th}$ century at the northern end to connect Schwerin with the Baltic Sea. Naturally, Schweriner See discharges towards the North Sea by the river Stör. During the construction of the Wallenstein trench in the 16$^{th}$ century, the major natural watershed between the Baltic Sea and the North Sea was cut through, which most likely changed discharge characteristics and might have led to a decline in lake level (Adolph et al., 2022; Carmer, 2006).

The regional climate in northern Germany is characterised by a gradient with decreasing temperature and precipitation from west to east influenced by the strength and direction of the Westerlies, which are controlled by the dominating mode of the NAO (Meinke et al., 2018). The climate at Schweriner See is affected by its distinct location in the transition zone from a more maritime to a more continental climate. For the period 1991-2020, the climate near the study site, as shown by data from the closest weather station Schwerin, was characterised by a warm-temperate climate with a mean annual temperature of 9.5 °C,

the coldest and warmest month being January (1.6 °C) and July (18.1 °C). Mean annual precipitation is 631 mm with dominating summer rainfalls (DWD Climate Data Center (CDC), 2022b, 2022c). The main wind direction is W to SSW (1967-2022, (DWD Climate Data Center (CDC), 2022a)), resulting in a fetch of 6-8 km for Schweriner Außensee.



# 3 Material and Methods

## 3.1 Coring and Composite Profile

Two parallel sediment cores, SAS21-11 and SAS21-12, were obtained in September 2021 from the deepest part of Schweriner See (52 m water depth, Fig. 1) using a 90-mm inner diameter UWITEC piston corer (www.uwitec.at). Additionally, a short sediment surface core (SAS22-2, 77.5 cm length) was retrieved in July 2022 using a UWITEC gravity corer (inner diameter: 60 mm) to guarantee an intact surface. All sediment cores were transported to the Physical Geography laboratory of the University of Greifswald and stored under dark and cool (~4 °C) conditions before further processing. Sediment cores were split, photographed, and sedimentological properties and sediment colour described according to standard protocols of the Physical Geography laboratory of the University of Greifswald. SAS22-2, SAS21-11 and SAS21-12 were spliced together using lithological marker layers, resulting in a composite sequence SAS21 of 1776.5 cm length. For this study, the upper 897.5 cm were investigated in detail.

## 3.2 Chronology

The chronology is based on 13 radiocarbon ages (Poznań Radiocarbon Laboratory) from terrestrial plant-macro fossils and 18 $^{210}$Pb/$^{137}$Cs ages in the uppermost part of the composite profile. $^{210}$Pb/$^{137}$Cs dating was carried out at the Environmental Radioactivity Research Centre of the University of Liverpool. Freeze-dried sediment samples from sediment core SAS22-2 were analysed for $^{210}$Pb, $^{226}$Ra, and $^{137}$Cs by direct gamma assay in the Liverpool University Environmental Radioactivity Laboratory using Ortec HPGe GWL series well-type coaxial low background intrinsic germanium detectors (Appleby et al., 1986). $^{210}$Pb was determined via its gamma emissions at 46.5 keV, and $^{226}$Ra by the 295 keV and 352 keV γ-rays emitted by its daughter isotope $^{214}$Pb following three weeks in storage in sealed containers to allow radioactive equilibration. $^{137}$Cs was measured by its emissions at 662 keV. The absolute efficiencies of the detectors were determined using calibrated sources and sediment samples of known activity. Corrections were made for the effect of self-absorption of low energy γ-rays within the sample (Appleby et al., 1992). Unsupported (fallout) $^{210}$Pb was calculated by subtracting $^{226}$Ra concentrations from the total $^{210}$Pb activities (Supplement S3-5). The age-depth model does not include the lowermost 210Pb/137Cs age, as only the upper 61 cm of sediment core SAS22-2 are part of the composite profile.

Except for this one age, all ages were used for age-depth modelling using the R-package 'rbacon' (v2.5.8, Blaauw and Christen, 2011) with the IntCal20 calibration dataset (Reimer et al., 2020) for calibration of radiocarbon data (Supplement S5-6). In the following, ages are reported as 'rbacon'-derived mean ages, including the upper and lower limits of the 95 % confidence interval (Fig. 2). The sedimentation rate was calculated based on this age-depth model.

## 3.3 Scanning techniques

Spectral analysis on sediment cores was carried out directly on the cling wrap covered freshly opened core surface using a Konica Minolta CM-2600d spectrophotometer (8 mm spot) in a 5 mm resolution. The spectral composition was recorded at





10 nm steps from 360 nm to 760 nm wavelength. Sediment core colour was calculated from L*a*b* provided by the SpectraMagic NX software (Konica Minolta) to RGB using the R-package farver (v2.1.1.9, Pedersen et al., 2022) and displayed on an age scale using Grapher (v20, Golden Software).

   Hyperspectral imaging was carried out at the Université Rouen Normandie. Measurements were performed on U-channels previously extracted from the cores in Greifswald using a VNIR-PDF hyperspectral camera (SPECIM) and subsequently
processed as described by Jacq et al. (2021) and van Exem et al. (2022). Images have a spatial resolution between 46x46 and 84x84 µm². Normalisation was carried out using the ENVI/IDL 5.5/8.2 software. Following van Exem et al. (2022), the spectral index $Area_{600-760}$ was used as an indicator for chloropigments indicating past *in-situ* productivity. To account for changes in average reflectance induced by changes in carbonate content, $Area_{600-750}$ was normalised with the $R_{mean}$.

   XRF-scanning was carried out at GEOPOLAR (Geomorphology and Polar Research) at the University of Bremen with an
XRF Core Scanner (ITRAX, Cox Analytics) at 2-mm step size with a Mo tube (30 kV, 50 mA, 5 s exposure time). Scanning XRF-derived elemental variations might be influenced by sample geometry, physical properties (e.g. water content, surface roughness, grain size variations) or scanner settings (Croudace and Rothwell, 2010; Weltje and Tjallingii, 2008). To reduce such effects, only elements with less than 5 % zero values (Si, K, Ca, Ti, Mn, Fe, Ni, Cu, Zn, Sr) were centre log-ratio (clr) transformed (Aitchison, 1982) using the PAST 4 software (Hammer, 2022). As proposed by Adolph et al. (2023) Cu, Ni and
Zn are used as a sum parameter $\sum(Cu,Ni,Zn)_{clr}$ for anthropogenic impact.

## 3.4 Sedimentological and Geochemical Analyses

   Discrete samples were taken in a 1 cm resolution using LL-channels (Nakagawa, 2014). Volume for dry-bulk density (DBD) was determined by the height and width of the LL-channels and sample resolution (V = 1 cm³ for SAS22-2 (upper 61 cm) and V = 3.24 cm³ for all other samples). DBD was calculated by dividing the dry weight of freeze-dried samples by the sample
volume. Loss-on-ignition (LOI) was determined on freeze-dried samples by heating the sediment to 550 °C for 3 h in a muffle furnace. Residues were used for grain size analysis and before processing for grain size analysis, a few samples were investigated using microscopic analyses to determine the sample composition. Carbonates were removed with 5 ml HCl (10 %) and samples were dispersed overnight in an overhead shaker with 5 ml sodium pyrophosphate. Measurements were carried out using a Laser Particle Sizer (Fritsch Analysette 22 microtec plus). The first reproducible of nine subsequent runs was used
for interpretation. Grain size statistics were calculated using the GRADISTAT 9.2 software (Blott and Pye, 2001).

   Carbonate content was determined on ground and homogenised samples by the Scheibler method on 0.17 to 0.55 g sample material. Subtracting carbonate content and $LOI_{550}$ from the total sample weight, the percentage of siliciclastics, which includes a share of silicious algae as revealed by microscopic analyses on the LOI ash residues, was calculated.

   Dried and homogenised sediment samples of 1.8 to 11.3 mg were used to analyse total carbon (TC) and total nitrogen (TN).
Concentrations were obtained using a Euro EA CNS analyser. TIC was determined with the IC Kit of the same device and TOC was calculated as TOC = TC – TIC. Measurements were calibrated against certified reference materials. Error estimates





are based on triple measurements of 18 samples. The precision is 0.77-5.25 % for TN, 0.24-0.89 % for TC and 0.68-19.19 % for TIC. The molar TOC/TN ratio was calculated based on molecular weights.

### 3.5 Leaf wax analyses

Leaf wax analyses were carried out at the Physical Geography department of the Friedrich-Schiller-University Jena. For this, two-centimetre-thick samples for leaf wax analyses were taken equivalent to a 100-150 years resolution and pooled with 0.5 cm of sediment above and below the sampling depth. Total lipids of the sediment samples (2.5 to 9.1 g dry sediment) were extracted with 40 ml dichloromethane (DCM) and methanol (MeOH) (9/1, v/v) using an ultrasonic bath over three 15 min cycles. The total lipid extract was separated by solid phase extraction using aminopropyl silica gel (Supelco, 45 μm) as the
stationary phase. The $n$-alkanes were eluted with 4 ml hexane and further purified using silver nitrate ($AgNO_3^-$; Supelco, 60-200 mesh). An Agilent 7890 gas chromatograph equipped with an Agilent HP5MS column (30 m, 320 μm, 0.25 μm film thickness) and a flame ionisation detector (GC-FID) was used for identification and quantification of the $n$-alkanes, relative to external $n$-alkane standards ($n$-alkane mix $n$-$C_{21}$ - $n$-$C_{40}$, Supelco).

Compound-specific stable hydrogen isotope analyses were carried out for the $n$-alkanes $C_{23}$ to $C_{31}$ using an IsoPrime vision
IRMS coupled to an Agilent 7890A GC via a GC5 interface operating in pyrolysis modus with a MaxChrome and silver wool-packed reactor at 1050 °C. The GC was equipped with a 30 m fused silica column (HP5-MS, 0.32 mm, 0.25 μm). Samples were injected splitless with a split–splitless injector and each sample was analysed in triplicate. $\delta^2H_{n\text{-}alkane}$ was measured against calibrated $H_2$ reference gas and all values are reported in per mille against VSMOW. The precision was checked by co-analysing a standard alkane mixture ($n$-$C_{27}$, $n$-$C_{29}$, $n$-$C_{33}$) with known isotope composition (Arndt Schimmelmann, University
of Indiana), injected in duplicate every nine runs. All measurements were corrected for drift, relative to the standard values in each sequence. $n$-$C_{23}$ to $n$-$C_{31}$ were abundant in sufficient amounts for compound-specific hydrogen analyses, but we will focus on $\delta^2H_{C25}$ in the following. Triplicates for the $\delta^2H_{C25}$ had a standard deviation of <3.3‰, the analytical error for the standard duplicates was <1.1‰ (n = 9). The $H3^+$ factor was checked every two days and stayed stable at 3.59 ± 0.08 (n= 3) during the measurements.

### 3.6 Pollen analyses

Altogether, 89 samples with a 1-2 $cm^3$ volume were used for pollen analysis. Samples were treated with 10 % hydrochloric acid (HCl) to dissolve carbonates, heated in 10 % potassium hydroxide (KOH) to remove humic compounds, and finally soaked in 40 % hydrofluoric acid (HF) for at least 24 h to remove the mineral fraction. Preparation was followed by acetolysis (Berglund and Ralska-Jasiewiczowa, 1986). One *Lycopodium* tablet (10679 spores; produced by Lund University) was added
to the samples (Stockmarr, 1971). Sample slides were analysed using an ECLIPSE 50i upright 130 microscope and counted to at least 500 arboreal pollen (AP) grains. Pollen taxa were identified using atlases (Beug, 2004; Moore et al., 1991) and the reference grains owned by the Institute of Geoecology and Geoinformation, Adam Mickiewicz University, Poznań. Non-Pollen Palynomorph Image Database was used to identify NPPs (Shumilovskikh et al., 2022). Pollen percentages were calculated





according to the formula: taxon percentage = (number of taxon grains/TPS) × 100%, where TPS indicates the total pollen sum

including the AP and non-arboreal pollen (NAP) taxa, and excluding the local and spore-producing plants and NPP taxa.

### 3.7 Diatom analyses

For diatom analysis, ~1 g of sediment was treated with HCl, $H_2O_2$, $H_2SO_4$ and $KMnO_4$ as described by Kalbe and Werner (1974). Residues were mounted on slides with Naphrax® to study them with a light microscope (Zeiss Axio Scope, oil-immersion Plan-Apochromatic objective, magnification 1000 X, numerical aperture 1.4). In total, 100 diatom samples were

counted and at least 450 diatom valves were counted for each sample. Diatom species identification and classification as eutraphentic diatoms followed Krammer and Lange-Bertalot (1988, 1986, 1991a, 1991b), Krammer (1997a, 1997b, 2000, 2002, 2003), Lange-Bertalot (2001) and Lange-Bertalot et al. (2017; 2011). The abundance of eutraphentic diatoms was calculated as proposed by Adolph et al. (2023).

### 3.8 Statistics

Similar sedimentological and geochemical composition intervals were established using a stratigraphically constrained cluster analysis on clr-transformed XRF data and sedimentological parameters. XRF data were scaled to a 1-cm resolution calculating the mean for each centimetre to account for differences in resolution and noise between XRF scans and sedimentological data. Calculations were carried out using the R package 'rioja' (v. 1.0.5) (Juggins, 2022). As the cluster analysis did not cover some changes or would have led to many clusters, we included an additional unit boundary based on visual inspection (Unit $C_1$ to

$C_2$). Pearson's r-values were calculated with the r-package 'Hmisc' (v. 5.0-1, Harrell Jr (2023)) (Supplement S1) and values with $p < 0.001$ are considered significant and mentioned in the text.




## 4 Results

### 4.1 Lithology, chronology and sedimentation rate

Based on the hierarchical constrained cluster analysis result, the 897.5 cm long sediment sequence was subdivided into six major lithological units (A-F, Fig. 2). Unit C was subdivided into $C_1$ and $C_2$ based on changes in $Ti_{clr}$ (Fig. 3) and D in three subunits ($D_1$-$D_3$) based on variations in organic matter variations (Fig. 2). Boundaries between units are mainly characterised by changes in organic matter reflected in sediment colour (Fig. 2) with lighter colours having an increased carbonate content and darker colour an increased organic matter content (e.g. Adolph et al., 2023; Strobel et al., 2022a; Wündsch et al., 2016;

Debret et al., 2011). Organic-rich sediment occurs from 878.5-844.5 cm sediment depth (Unit B) and, similarly, in Unit $D_2$ (Fig. 2). In contrast, carbonate content is the highest in Unit C. Otherwise, the sediment is composed of siliciclastic material and a share of diatoms and other silicious algae, which somewhat increase above 752.5 cm sediment depth marking the boundary between unit $C_1$ and $C_2$ (Fig. 2).

Bayesian age-depth modelling gave a mean age of $3070^{+170}/_{-210}$ cal BP for the bottommost sample considered for interpretation

in this contribution (897.5 cm). All ages are in stratigraphic order and overlap with the 95 % confidence interval of the age-depth model (Fig. 2). The top of the composite profile is determined by the recovery of the gravity core forming the top of the sequence (July 2022). Total $^{210}$Pb activity reached values close to equilibrium at 65 cm sediment depth. Concentration of the artificial radionuclide $^{137}$Cs has a well-defined peak at 29-28 cm suggesting that this peak records fallout from the 1986 Chernobyl accident (Fig. 2). A smaller and less distinct peak at 45-44 cm may record the early 1960s fallout peak from the

atmospheric testing of nuclear weapons. The well-resolved Chernobyl $^{137}$Cs peak suggests relatively little sediment mixing within this core. The sedimentation rate is 2-3 mm a$^{-1}$ between 897.5 cm ($3070^{+170}/_{-210}$ cal BP) and 298 cm ($620^{+35}/_{-50}$ cal BP) (Fig. 2) and increases to 4 mm a$^{-1}$ at 56 cm ($7^{+10}/_{-10}$ cal BP). Above the record yields a much higher sedimentation rate of 5-10 mm a$^{-1}$.

### 4.2 Geochemical and Paleontological Analyses

#### 4.2.1 Correlations

$LOI_{550}$, TOC, TN and inc/coh are significantly correlated (r > 0.70, Supplement S1) and agree visually well with $Area_{600-760}$ (Fig. 3). Since $Area_{600-760}$ reflects *in-situ* chloropigments (e.g. van Exem et al., 2022), we consider $LOI_{550}$, TOC, TN and inc/coh as indicative for in-lake productivity, which is supported by TOC/TN values mostly < 12 indicative for a dominance of nonvascular aquatic plants with only a small contribution of vascular plants (Meyers and Ishiwatari, 1993). Moreover, the

individual and summed ($C_{21}$-$C_{35}$) *n*-alkane concentrations correlate with the in-lake productivity of Schweriner See (r > 0.8, Supplement S2), indicating a predominance of in-situ aquatically-derived *n*-alkanes (e.g. Strobel et al., 2022b; Sachse et al., 2004). The compound-specific isotopic hydrogen signatures ($\delta^2$H) of the *n*-alkanes $C_{23}$ to $C_{31}$ show a comparable pattern (Supplement S2), further indicating a predominantly aquatic origin of the *n*-alkanes (e.g. Strobel et al., 2022b; Sachse et al., 2004).



Ca, Sr, TIC, and Sr/Ca indicate carbonate precipitation (Haberzettl et al., 2005; Haberzettl et al., 2019; Haberzettl et al., 2009). They are significantly negatively correlated to productivity indicators (r > -0.76), suggesting that one parameter dilutes the other. As biogenically precipitated calcite has higher Sr contents than inorganically precipitated calcite (Hodell et al., 2008), Sr/Ca suggests changes in the carbonate precipitation mechanism. The Schweriner See record supports this by significantly correlating Sr/Ca with productivity indicators (e.g. inc/coh, r = 0.80, $LOI_{550}$, r = 0.62). The fact that higher Sr/Ca ratios occur

during phases of increased productivity indicators leads to the conclusion that productivity is driving the dilution mechanism described above, diluting carbonate precipitation. Moreover, it suggests that carbonate precipitation might be the background sedimentation in Schweriner See diluted by changes in productivity.

Indicators for minerogenic input ($Ti_{clr}$ and $K_{clr}$, r =0 .79, Davies et al., 2015; Haberzettl et al., 2005) show a significant positive correlation. Often, Ti is associated with silty sediments (e.g. Davies et al., 2015; Kylander et al., 2011), but here, $Ti_{clr}$ is in

good agreement with mean grain size (r = 0.54), which is mostly related to variations in sand content (r = 0.94).

In the following, these parameters will only be addressed by their underlying processes, i.e. productivity, carbonate precipitation and minerogenic input. However, some parameters cannot be assigned to one of the groups. TOC/TN, for example, shows no strong significant correlations to any parameter. $Fe_{clr}$ is strongly positively correlated to $LOI_{550}$ (r = 0.83). However, one would rather expect a negative correlation due to an oxygen deficit by degradation processes in organic-rich

layers, given the correlation of r = 0.66 of Fe to Mn. Alternatively, a correlation to minerogenic input would be plausible, which is not the case (Ti: r = 0.21). As the interpretation of Fe is not straightforward and needs further investigation, Fe will not be considered in the following. $\sum(Cu,Ni,Zn)_{clr}$ is correlated to eutraphentic diatoms (r = 0.63), and according to previous studies (Adolph et al., 2023) both reflect human impact.

### 4.2.2 Zonation

Although this study focuses on the well-dated upper 897.5 cm of the composite record SAS21, to put the results in a broader perspective, parameters are shown down to 1000 cm (Fig. 3). The section between 1000-897.5 cm is characterized by continuously increasing productivity and minerogenic input from 1000 to 916.5 cm above which productivity decreases again (Fig. 3). Arboreal pollen (AP) is around 94 %. Planktonic diatoms vary between 65-83 % and eutraphentic diatoms are low (Fig. 3). At 897.5 cm, AP decrease to 74.7 % marking the starting point of the well-dated part of the record, for which leaf-

wax *n*-alkanes were also analysed. This section covers the last $3070^{+170}/_{-210}$ cal BP years (Fig. 4) and is characterized as follows: **Unit A (897.5-878.5 cm)** is characterized by rapidly increasing productivity (e.g. $LOI_{550}$ from 17.5 % to 47.5 %, Fig. 3). The *n*-alkane concentration is 19 µg $g^{-1}$ and $\delta^2H_{C25}$ -158.2 ± 0.5 ‰. AP pollen is ~93 %. In **Unit B (878.5-844.5 cm)**, the minerogenic input (e.g. $Ti_{clr}$) increases with a maximum at 875.5 cm, which also marks maximum productivity (e.g. 62.3 % $LOI_{550}$) and minimum carbonate precipitation (e.g. 8.9 % carbonates). Above 844.5 cm, productivity and minerogenic input

indicators continuously decrease. $\sum(Cu,Ni,Zn)_{clr}$ is increased and AP pollen is ~90 %. The abundance of eutraphentic diatoms is overall low and planktonic diatoms are high. A decline in productivity co-occurs with more negative $\delta^2H_{C25}$ from -159.8 ± 0.2 to -155.0 ± 0.5 ‰. **Unit C (844.5-691.5 cm)**, which is subdivided into **Unit C₁ (844.5-752.5 cm)** and **Unit C₂**





**(752.5-691.5 cm),** is characterised by low productivity, while $\delta^2 H_{C25}$ fluctuates but remains more negative compared to previous units with values as low as -166.4 ± 0.7 ‰ and *n*-alkane concentration between 7.5-18.8 µg g$^{-1}$. The diatom species *Stephanocostis chantaicus* Genkal & Kuzmina occurs for the first time in Unit $C_1$. TOC/TN increases from 10.5 to 17.6 and declines to 8-10 in Unit $C_2$. Unit $C_2$ is characterized by a steep increase in minerogenic input (e.g. $Ti_{clr}$ and $K_{clr}$) with two maxima. It subsequently decreases but remains on an overall higher level. Pollen composition has ~90 % AP with a maximum of 98.9 % at 788.5 cm. The abundance of eutraphentic diatoms is ~12 % and planktonic diatoms are common, reaching up to 52.7 %. **Unit D (691.5-368.5 cm)** is subdivided into **D$_1$ (691.5-582 cm), D$_2$ (582-429 cm)** and **D$_3$ (429-368.5 cm)**. The diatom species *S. chantaicus* disappears completely in unit D (Fig. 4). The TOC/TN ratio is around 9-11 and minerogenic input indicators have a local minimum at 602 cm. The same local minimum is also reflected in productivity indicators, e.g. an $LOI_{550}$ of 16.4 % compared to 36.7 % at 618 cm. The abundance of planktonic diatoms increases continuously to 73 %. AP continuously increases concurrent with a change in forest composition by strongly increasing *Fagus sylvatica* and *Carpinus betulus* percentages. This unit is characterized by high productivity (e.g. 63.5 % $LOI_{550}$) and low carbonate precipitation (e.g. 6.8 % carbonates). $\delta^2 H_{C25}$ are more positive in Unit $D_1$ compared to Unit C with values from -163.0 ± 1.4 to -152.7 ± 1.3 ‰, which marks the most positive value of the record at 525.5 cm. Minerogenic input has several peaks. Unit $D_2$ marks an overall change in the diatom assemblage as the share of planktonic diatoms increases until it remains > 80 % above 453.5 cm. Unit $D_3$ shows an overall decrease in productivity signals simultaneous to increased carbonates. $\delta^2 H_{C25}$ is more negative again and AP remains high at ~91 %. A decline in productivity is observed at the boundary to **Unit E (368.5-109.5 cm)** with $LOI_{550}$ values dropping from around 32 % to 10.9 % at the top of unit $D_3$ at 305.5 cm. $\delta^2 H_{C25}$ gets continuously more negative to -163.1 ± 3.3 ‰ at 263 cm; however, values fluctuate within this unit. Similar to Unit C, lower productivity concurs with the occurrence of *S. chantaicus*. $\sum(Cu,Ni,Zn)_{clr}$ shows increased but fluctuating values again, which correlate with a drop in AP to 82 % with contemporaneous vegetation changes from a declining dominance of *F. sylvatica* and *C. betulus*. **Unit F (109.5-0 cm)** marks a clear shift, which is especially reflected in several productivity parameters such as $Area_{600-760}$, inc/coh and the abundance of eutraphentic diatoms, which was around 12 % in all previous units but increases to 92.7 % in unit F.



# 5 Discussion

Earlier studies showed that close-by but small lake Rugensee (~5 km west of the coring location, 55 ha) responded sensitively to settlement activities since the Neolithic with an increased trophy (Dreßler and Hübener, 2011). In contrast, Schweriner See

shows oligotrophic to mesotrophic conditions from the bottom of the record to $105^{+95}/_{-75}$ cal BP (=$1845^{+75}/_{-95}$ CE, Fig. 4). A considerable local anthropogenic impact on the trophic state is only evident after that (Unit F) confirming the assumption that larger systems are less susceptible to anthropogenic impact. In contrast, smaller lakes suffer from an anthropogenic overprinting of the climate signal. Adolph et al. (2023) previously discussed the period of anthropogenic influences after $105^{+95}/_{-75}$ cal BP (=$1845^{+75}/_{-95}$ CE), identifying sewage discharge and catchment population density as the main drivers for high

lake eutrophication and contamination.

## 5.1 Hydroclimatic variations influenced by NAO polarity

### 5.1.1 Productivity and δ2H as Indicators for NAO-related hydroclimatic variability

In-lake productivity, exemplified by the inc/coh ratio in Fig. 4 as it has the highest resolution, is often interpreted as either controlled by temperature and/or nutrient input (Kasper et al., 2013; Günther et al., 2016; Doberschütz et al., 2014). As the

abundance of eutraphentic diatoms suggests an increase in eutrophication only after $105^{+95}/_{-75}$ cal BP ($1845^{+75}/_{-95}$ CE) (Unit F, Fig. 4), productivity was likely driven by temperature variability before. Distinct temperature changes are supported by the recurrent occurrence of the diatom species *S. chantaicus,* which is associated with long-lasting ice covers (Scheffler and Padisák, 2000) concurrent with low productivity phases (Fig. 4). Such prolonged ice covers indicate lower winter temperatures and longer lake ice-cover duration, which can substantially affect the seasonal heat budget, timing and length of stratification

but also the productivity of aquatic ecosystems (e.g. Bonsal et al., 2006). Therefore, we conclude that before $105^{+95}/_{-75}$ cal BP in-lake productivity was mainly driven by winter temperature variability modulating ice cover duration, heat budget and growing season length, most likely modulated by large-scale atmospheric patterns (e.g. Schmidt et al., 2019; Bonsal et al., 2006; Blenckner et al., 2007). Similar observations of winter temperatures influencing in-lake productivity by modulating ice-cover duration and growing season length have been made for a lake in western-central Sweden (Czymzik et al., 2023).

Regarding *n*-alkanes, lacustrine sediments generally contain a mixed signal from terrestrial and aquatic sources, which can be distinguished by their chain-length distribution (e.g. Strobel et al., 2021; Ficken et al., 2000). Classically, long-chain *n*-alkanes (e.g., $C_{27}$-$C_{31}$) are suggested to be produced as leaf waxes by higher terrestrial plants and primarily incorporate the local growing season precipitation as their primary source water for photosynthesis (e.g. Sachse et al., 2012; Strobel et al., 2020; Strobel et al., 2022a). However, the $\delta^2H$ signal of precipitation mainly depends on the atmospheric moisture source of the

precipitation in the mid-latitudes (Strobel et al., 2020; Strobel et al., 2022b; Bliedtner et al., 2020; Wirth and Sessions, 2016) and also additional fractionation processes can occur at the plant-soil interface, with evaporation of soil water and transpiration of leaf water being prominent factors (Feakins and Sessions, 2010; Kahmen et al., 2013; Zech et al., 2015). In contrast, short-chain *n*-alkanes are produced by aquatic macrophytes and algae (e.g., $C_{21}$-$C_{25}$) and incorporate the $\delta^2H$ signal of the lake's





water, which integrates the $\delta^2$H precipitation signal throughout the year. Depending on the morphometric and hydrological
parameters of the lake itself, lake water can be strongly modulated by evaporative lake water enrichment (e.g. Aichner et al.,
2022; Mügler et al., 2008; Sachse et al., 2004; Strobel et al., 2022a). Notably, this classic *n*-alkane source attribution (terrestrial
vs aquatic) is not always trivial because, for example, aquatic emergent plants can also synthesize distinct amounts of long-
chain *n*-alkanes ($\geq$C$_{27}$), which then also incorporate the $\delta^2$H signal of the lake's water, challenging the interpretation of the $\delta^2$H
signal (Ficken et al., 2000; Yang and Bowen, 2022)

At Schweriner See, *n*-alkane concentration, as well as individual *n*-alkanes and their respective $\delta^2$H signals, are significantly
correlated with in-lake productivity indices (Fig. 3; Supplement S1-S2) and we therefore suggest that the majority of the
*n*-alkanes is of aquatic origin. Although the compound-specific $\delta^2$H of all detectable *n*-alkanes shows a comparable pattern,
mixing can complicate the interpretation of the longer-chained *n*-alkanes and we will therefore focus on $\delta^2$H of C$_{25}$ ($\delta^2$H$_{C25}$) in
the following because C$_{25}$ and its $\delta^2$H signal provide the most robust aquatic end-member. $\delta^2$H$_{C25}$ is more enriched during
periods of higher (milder) winter temperatures and more depleted during periods of colder winter temperatures (Fig. 4), which
can be due to the following two explanations: i.) Since the aquatically-derived $\delta^2$H$_{C25}$ primarily reflects $\delta^2$H of the lake's water
and year-round precipitation, Schweriner See's position in the mid-latitudes suggests that $\delta^2$H$_{C25}$ is mostly related to moisture
source changes in the North Atlantic region. More enriched $\delta^2$H$_{C25}$ values may correspond to isotopically enriched
southern/central North Atlantic precipitation sources. In contrast, more depleted $\delta^2$H$_{C25}$ values originate from isotopically
depleted precipitation from the northern North Atlantic and/or Arctic region. On the other hand, ii.) enriched $\delta^2$H$_{C25}$ could also
result from temperature-driven evaporative enrichment of the lake water, as frequently reported from semi-arid regions
(Mügler et al., 2008; Strobel et al., 2022a). Such observations have also been made for several smaller lakes from northeastern
Germany (Aichner et al., 2022). However, these lakes are located ca. 120 km southeast of Schweriner See in the more
continental climate zone where evaporative enrichment may have an increased influence on the isotopic composition of lake
water.

Such distinct variations in winter surface air temperatures, precipitation sources, and/or evaporative enrichment are mainly
modulated by the North Atlantic Oscillation (NAO) in the North Atlantic area (Hurrell and Deser, 2009). The barometric
difference between high- and low-pressure systems over the Azores and Iceland affects Westerly strength and pathways and
eventually the moisture source for northern Central Europe. Milder winter temperatures are associated with NAO+ initiated
by strong high- and low-pressure systems over the Azores and Iceland resulting in strong Westerlies, which bring moist and
mild air from the southern North Atlantic with more positive $\delta^2$H values to northern Central Europe (e.g. Breitenbach et al.,
2019; Hurrell, 1995; McDermott et al., 2011; Baldini et al., 2008; Comas-Bru et al., 2016). In contrast, during NAO-
conditions, pressure systems are weakened, which allows frequent atmospheric blocking redirecting the Westerlies southward
and a frequent intrusion of cold and dry air from northern North Atlantic and Arctic regions, which is associated with more
negative $\delta^2$H values (e.g. Breitenbach et al., 2019; Hurrell, 1995; McDermott et al., 2011; Baldini et al., 2008; Comas-Bru et
al., 2016). Still, changes in winter temperature may at least partly drive evaporative enrichments with higher enrichment during





milder winter temperature phases, reflecting a positive NAO and, conversely, lower enrichment coinciding with colder winter temperatures linked to a negative NAO.

Considering this interpretation, from $3030^{+170}/_{-210}$-$2820^{+180}/_{-180}$ cal BP and $2110^{+160}/_{-130}$-$830^{+100}/_{-90}$ cal BP, milder winter temperatures (inc/coh, Fig. 5) and more enriched $\delta^2H_{C25}$ are interpreted as NAO+ conditions. Northward displaced westerlies bring isotopically enriched precipitation from the southern/central North Atlantic to northern Central Europe and/or warmer temperatures may result in a higher evaporative environment. Conversely, from $2820^{+180}/_{-180}$-$2110^{+160}/_{-130}$ cal BP and $830^{+100}/_{-90}$-$105^{+95}/_{-75}$ cal BP, colder winter temperatures with prolonged ice cover duration and more depleted $\delta^2H_{C25}$ values correspond to negative NAO phases. Westerlies are displaced southward and a more frequent atmospheric blocking allows for the intrusion of northerly winds with precipitation from the northern Atlantic and Arctic region and/or colder temperatures (Fig. 5). Rates of changes between positive to negative conditions vary between the individual phases e.g. with a rapid drop in winter temperature (inc/coh) around $2820^{+180}/_{-180}$ cal BP but an gradual increase from $2110^{+160}/_{-130}$-$1720^{+70}/_{-70}$ cal BP (Fig. 5).

### 5.1.2 NAO variability during the past 3 ka on an interregional scale

A comparison to other NAO-sensitive records from Norway, Scotland, Sweden and Germany (e.g. Faust et al., 2016; Baker et al., 2015; St. Amour et al., 2010; Breitenbach et al., 2019; Waltgenbach et al., 2021; Olsen et al., 2012) reflecting past NAO variations shows a good agreement with signals from Schweriner See (Fig. 5). Moreover, data from the precipitation sensitive record from Dosenmoor (Barber et al., 2004; Charman et al., 2009, Fig. 5) aligns well with changes in the moisture source region. As expected under a positive NAO influence, a southern moisture source region causes wetter conditions. In comparison, a northern moisture source region under negative NAO conditions causes drier conditions (Fig. 5) due to shifts in the westerly pathway.

Similar to our record, positive NAO conditions for $3030^{+170}/_{-210}$-$2820^{+180}/_{-180}$ cal BP were inferred for Central Scandinavia (St. Amour et al., 2010) and Greenland (Olsen et al., 2012) (Fig. 5). Subsequent negative NAO conditions from $2820^{+180}/_{-180}$-$2110^{+160}/_{-130}$ cal BP are also in accordance with predominantly negative NAO conditions reconstructed from various other records (Olsen et al., 2012; Becker et al., 2020; Faust et al., 2016; Baker et al., 2015; St. Amour et al., 2010) (Fig. 5). In addition to this negative NAO conditions, for the North Atlantic region, a shift to cooler conditions occurs around 2800 cal BP (2.8 ka event) attributed to changes in solar activity (Homeric Grand Solar minimum, ~2800–2550 cal. BP, Reimer et al., 2020). The onset of the Homeric Grand Solar minimum is within the error range of observed cooler conditions ($2820^{+180}/_{-180}$ cal BP) at Schweriner See. These changes in solar activity triggered a rapid climate change and likely changes in atmospheric circulation patterns, which are linked to cooler and/or wetter and/or windier conditions (e.g. Engels et al., 2016; Martin-Puertas et al., 2012; Rach et al., 2017; van Geel et al., 2014; van Geel et al., 2000; Mellström et al., 2015; Harding et al., 2023; Martínez Cortizas et al., 2021). Some studies associate solar minima with shifts to a negative NAO (e.g. Shindell et al., 2001; Gray et al., 2016), as observed in this study at Schweriner See. Other studies suggest a weakening of the subpolar gyre, resulting in changes in the atmospheric circulation, for example, by more frequent and persistent atmospheric blocking (Moffa-Sánchez et al., 2014), as observed under negative NAO conditions. Sjolte et al. (2018) suggest a complex response to solar minima, which





is not directly linked to NAO but rather to the Eastern Atlantic pattern with increased mid-Atlantic blocking and shifts to intensifying northerly winds resembling negative NAO conditions. Shifts in the Eastern Atlantic pattern during Grand Solar Minima are supported by Harding et al. (2023) for the North Sea region.

For Schweriner See, a more northern moisture source region and/or low temperature driving evaporative lake water enrichment is inferred until $2110^{+155}/_{-130}$ cal BP, indicating prevailing negative NAO conditions beyond the Grand Solar Minima. A similar

phenomenon with cooler conditions for 2800-1650 cal BP was also observed at close-by Rugensee (Dreßler et al., 2011). Contemporaneously, for 2550-2050 BP (OSL), dominating northerly to easterly winds are reported (Lampe and Lampe, 2018) for the close-by Darss area (ca. 110 km northeast of Schweriner See), which are also commonly associated with negative NAO conditions observed in SAS21.

The shift to warmer positive NAO conditions from $2110^{+160}/_{-130}$-$830^{+100}/_{-90}$ cal BP with a gradual increase in winter temperature

until $1720^{+70}/_{-70}$ cal BP coincides with the Roman Warm Period (RWP, c. 2150-1550 cal BP), which was a period of general warmth and dryness in Europe (Lamb, 2013). Similarly, shifts to positive NAO conditions have been reconstructed from different archives from Scotland, Norway and Central Scandinavia (Fig. 5, St. Amour et al., 2010; Baker et al., 2015; Faust et al., 2016) and considering chronological uncertainties, it is in accordance with NAO reconstructions from Greenland (Olsen et al., 2012) suggesting predominantly stable positive NAO circulation pattern from 2000-550 cal BP. Contemporaneously, a

change in forest composition occurs (Fig. 4), most likely induced by milder and moister winter conditions leading to optimal climatic conditions for the expansion of beech (*Fagus sylvatica*) and hornbeam (*Carpinus betulus*) as both maxima coincide with higher winter temperatures (Fig. 3, Fig. 4, e.g. Bradshaw et al., 2010). However, anthropogenic activities, e.g., soil changes, cannot be excluded from these species' spread(Giesecke et al., 2017).

Predominantly negative NAO conditions between $830^{+100}/_{-90}$-$105^{+95}/_{-75}$ cal BP are contemporaneous with a long-term cooling

trend associated with repeated phases of volcanic-solar downturns in Europe (PAGES 2k Consortium, 2013). Compared to the previous negative NAO phase, which coincides with lower solar activity, this period shows a stable low winter temperature but repeated shifts to a northern moisture source region and/or low temperature driven lake water evaporative enrichment, e.g. around $860^{+95}/_{-95}$ and $540^{+65}/_{-90}$ cal BP. Considering chronological uncertainties, both shifts might also align again to Grand Solar Minimas, i.e. the Oort (940-880 cal BP) and Spörer (560-400 cal BP) solar minima (Usoskin et al., 2007). From 800-

500 BP (OSL), the negative NAO conditions are confirmed by frequent strong winds from northern and eastern directions (Lampe and Lampe, 2018).

## 5.2 Minerogenic input as an indicator for various interacting processes

### 5.2.1 Processes affecting minerogenic input

Minerogenic elements titanium and potassium are often regarded as a proxy for minerogenic input from the catchment

(Haberzettl et al., 2005; Haberzettl et al., 2019) associated with two main interpretations, i.e. windier and/or wetter conditions (Davies et al., 2015). Normally, one would expect that under increased windiness, the minerogenic input increases because an





additional aeolian component would be introduced to the lake. However, this is unlikely for our record because Schweriner Außensee is surrounded by a cliff on the western shoreline (Fig. 1) serving as a wind shelter, and pollen composition suggests a closed canopy forest (Fig. 4) inhibiting aeolian erosion and transport. Moreover, under wetter conditions, one would expect

an increased minerogenic input because an increased surface run-off would bring more allochthonous material into the lake (Haberzettl et al., 2007). However, Schweriner See has hardly any above-ground inflow and is mainly fed by groundwater, which has no impact on particulate minerogenic matter transport. Therefore, wetter conditions result in higher lake levels but without a minerogenic matter supply to the coring location. Higher lake levels are, for example, indicated by paleo-lacustrine landforms (e.g. beach ridges, nearshore bar) from the north-eastern shoreline of Schweriner See (Adolph et al., 2022) for $3020$

$\pm 260$, $330 \pm 50$ and $260 \pm 40$ BP (OSL). However, such high lake levels coincide with lower minerogenic input in SAS21. In contrast, phases of lower lake levels are implied for 1050-950 BP (archaeological findings, Konze, 2017; Lorenz et al., 2017), $585 \pm 75$ BP (OSL, Adolph et al., 2022), and 120-100 BP (historical documents, Umweltministerium Mecklenburg-Vorpommern, 2003) coinciding with a higher minerogenic input to SAS21 (Fig. 5).

As aeolian input and above-ground inflow are of minor importance for Schweriner Außensee (Wöbbecke et al., 2003), we

suggest that minerogenic input is mainly modulated by the unique morphometry of the lake basin, which is characterized by a broad, shallow water area in front of the eastern shoreline (Fig. 1B). We assume this area as a main source for minerogenic material as surface sediment sampling revealed highest values for minerogenic elements there (e.g. Ti, K, Adolph et al., 2023). This area is highly susceptible to wind-induced wave action as it is exposed to the main wind direction with a fetch of 6-8 km. During higher (lower) lake levels, the shallow water area would be further away (closer) from the coring site, which results in

a reduced (higher) transport of wave-eroded minerogenic material towards the coring site. Further support for this interpretation comes from the good correlation of minerogenic elements K and Ti to grain size mean. Both grain sizes mean or medians have previously been used in large lakes as a paleo-shoreline distance indicator, e.g. Kasper et al. (2012) arguing that during episodes of higher lake levels – and therefore a larger paleo-shoreline distance of the coring location – coarser grains did not reach the coring location. Similar suggestions have been made by Bonk et al. (2023) for Lake Lubińskie, where

under lower water levels, shorelines were exposed and more susceptible to erosion and, consequently, Ti and quartz grains increased at the coring location. In the following, $Ti_{clr}$, which has a lower signal-to-noise ratio than K, will be used as minerogenic input indicator mainly depending on paleo-shoreline distance related to lake level changes (Fig. 5).

As a second factor, minerogenic input is also controlled by wind-induced wave action, wind speed and direction changes, which might have affected proxy sensitivity. Wind speed and storminess changes probably influenced minerogenic input as

this controls wave energy and, consequently, the amount of material eroded and transported. Therefore, wind activity might sometimes dominate the Ti signal instead of the lake level. One of these instances when wind speed and storminess on Schweriner See dominated Ti deposition at SAS21 could be at $3020 \pm 260$ BP (OSL) when a nearshore bar investigated by Adolph et al. (2022), indicates both a high lake-level phase with windy conditions. Several layers of very coarse grains (> 2 mm) were deposited within this nearshore bar, which is only possible under high wave energy driven by increased wind

speed. In accordance, increased storminess was suggested for the Danish North Sea coast between 3300-2800 BP (Goslin et



al., 2018) and SW Sweden from 3050-2850 BP (Björck and Clemmensen, 2004). Therefore, the concurrently increased minerogenic input ($3020^{+180}/_{-210}$-$2940^{+190}/_{-200}$ cal BP, Fig. 5) is likely related to strongly increased wind-induced wave energy. In contrast to storminess affecting the minerogenic input, an actual lower lake level for this period is supported by drier conditions at Dosenmoor during the nearshore bar deposition (Fig. 5), supporting our initial paleo-shoreline proximity

interpretation. However, the sediment sequence of the nearshore bar suggests continuous sedimentation with no evidence of post-depositional erosion (Adolph et al., 2022). This inconsistency could be resolved if a drop in lake level concurrent with stormier conditions is assumed. Such a drop may lead to the deposition and preservation of the nearshore bar and a higher Ti input due to both processes, i.e. windier conditions and a subsequent lower lake level.

The interpretation of minerogenic input as a relative lake-level indicator is based on today's prevailing wind direction from
SW to W, which results in a fetch of 6-8 km. However, changing wind directions to more northerly and easterly wind directions might have also influenced the erosional processes. Consequently, minerogenic input to the coring location of SAS21 at Schweriner See would be reduced because the coring location is closer to the western than eastern shoreline (Fig. 1). Such a change in wind direction was, for example, reconstructed for the close-by Darss area (Lampe and Lampe, 2018). Under prevailing northerly or easterly winds, wave action would have increased at the western shoreline, which might have increased
minerogenic input, suggesting a lower lake level even if only the wind direction changed. However, northerly to easterly winds are associated with drier conditions in NE Germany. Therefore, minerogenic input might reflect a lower lake level due to drier conditions and a decreased shoreline distance due to changes in the prevailing wind direction.

To test the reliability of $Ti_{clr}$ as a proxy for shoreline distance (i.e. lake-level variations), it is compared bog surface wetness reconstructions from peat bog Dosenmoor (ca. 105 km northwest of Schweriner See, Fig. 5, Barber et al., 2004; Daley and
Barber, 2012) both mirroring moisture availability in the following. Bog surface wetness is assumed to reflect the summer moisture deficits mainly driven by precipitation but reinforced by temperature (Charman et al., 2009). Similar processes affecting the lake level at Schweriner See have been observed recently. For example, a summer moisture deficit due to prevailing dry conditions in 2018 resulted in a severe lake-level decline (Landesamt für Umwelt, Naturschutz und Geologie Mecklenburg-Vorpommern, 2018), which could not be completely compensated by winter precipitation. Therefore, we suggest
that bog surface wetness is a suitable proxy for comparison. For the past 3000 years, higher and lower lake level phases derived from Schweriner See align well with reconstructed wetter and drier conditions at Dosenmoor (Fig. 5, Barber et al., 2004) in all but one instances, i.e. the period from $1660^{+40}/_{-50}$-$1120^{+90}/_{-100}$ cal BP. For this period, a lower lake level is suggested by $Ti_{clr}$ (Fig. 5), indicating drier conditions, which contrasts with hydroclimatic reconstructions from Dosenmoor (Daley and Barber, 2012), other records (Magny, 2004; Büntgen et al., 2021; Starkel et al., 2013, Fig. 6) and our NAO reconstruction implying a
positive NAO associated with milder and wetter conditions at the same time (Fig. 5). However, positive NAO conditions are also associated with an enhanced storm activity (e.g. Hurrell et al., 2003), which is in accordance with stormier conditions during the period of interest in Northwest Europe from 1700-1100 cal BP (Pouzet et al., 2018) and 1900-1050 cal BP (Sorrel et al., 2012) and in southwestern Sweden around 1500 cal BP (Jong et al., 2007; Jong et al., 2006). Therefore, we assume increased storminess masked our lake-level signal for this period.




In conclusion, the main driver for minerogenic input to the coring location of SAS21 at Schweriner See were lake-level variations with additional wind speed influences and direction amplifying wave action. Consequently, the lake level was higher than today for $3070^{+170}/_{-210}$-$2380^{+170}/_{-150}$ cal BP. Afterwards, the lake level was lower for $2380^{+170}/_{-150}$-$2050^{+130}/_{-110}$ cal BP before it rose again until $1660^{+40}/_{-50}$ cal BP. For $1660^{+40}/_{-50}$ cal BP-$1120^{+90}/_{-100}$ cal BP, the lake-level signal was most likely masked by increased storminess and might have been higher than today. A lower lake level for $1050^{+90}/_{-70}$-$850^{+100}/_{-90}$ cal BP,

which aligns with a suggested lake level of at least 2 m below today's for the same period (Konze, 2017; Lorenz et al., 2017). This phase is followed by a higher lake level from $850^{+100}/_{-90}$-$650^{+40}/_{-40}$ cal BP and a lower lake level from $650^{+40}/_{-40}$-$410^{+95}/_{-110}$ cal BP, which coincides with peat deposits below today's lake level around $530^{+35}/_{-25}$ cal BP (Adolph et al., 2022). A higher lake level is indicated for $410^{+95}/_{-110}$-$210^{+105}/_{-95}$ cal BP supported by two beach ridge deposits dated $330 \pm 50$ and $260 \pm 40$ BP (OSL). The subsequent lake-level decline concurs with the construction of the so-called Wallensteingraben in the 16[th] century,

which was built to connect Schweriner See with the Baltic Sea. By establishing a second outflow, the outflow regime was likely changed (Carmer, 2006; Adolph et al., 2022). The expansion of the Stör waterway in the mid-19[th] century also resulted in a lower lake level (Fellner, 2007; Umweltministerium Mecklenburg-Vorpommern, 2003), which resulted in the division into two lake basins we see today (Fig. 1).

### 5.2.2 Late Holocene regional lake-level variations

As Schweriner See has been influenced by NAO variability, most likely, regional lake-level variations have also at least partly been driven by large-scale atmospheric North-Atlantic variations. Lower lake levels generally align with a northern moisture source region (Fig. 5) associated with drier winters, which indicates that winter precipitation may have a noticeable influence on lake-level variations. This is in accordance with modelling approaches by Vassiljev (1998), who suggested that lakes in temperate humid areas are likely highly sensitive to changes in winter precipitation and that lake-level changes from lakes,

which are large in comparison to their catchment like Schweriner See, are more likely to reflect changes in precipitation.

The influence of large-scale atmospheric changes on lake-level variability during the past 3000 years explains the similar patterns observed in different archives (e.g. lacustrine sediments, peat bogs, tree rings) reflecting lake-level variations and hydroclimatic conditions in Denmark (Barber et al., 2004), NE-Germany (Daley and Barber, 2012; Theuerkauf et al., 2022), western Poland (Pleskot et al., 2018; Bonk et al., 2023; Starkel et al., 2013) but also Eastern Central Europe (Büntgen et al.,

2021) and the Jura mountains (Magny, 2004) (Fig. 6). Offsets might occur due to chronological uncertainties or proxy sensitivity in some areas.

However, additional supraregional drivers may have affected lake-level variability as well. For example, predominantly higher lake levels were reconstructed for most of the archives listed above, including Schweriner See (Fig. 6) from 2800-2500 cal BP. Those are most likely linked to prevailing colder and/or wetter conditions during the Homeric Grand Solar Minimum in

northern Germany (e.g. Barber et al., 2004; Theuerkauf et al., 2022). Such wetter conditions contradict the reconstructed more negative NAO conditions associated with cold and dry winter conditions. This indicates additional influences such as changed summer precipitation and/or evapotranspiration or an additional influence of the Eastern Atlantic pattern during solar minima



(Sjolte et al., 2018). Though NAO is more active in the cold season (October – April) with larger amplitudes and a strong influence on winter temperature and precipitation (Hurrell et al., 2003), it does not explain summer temperature and precipitation variability. Therefore, solar activity has been suggested as one key driver for Holocene climatic variability in the Jura mountains, where higher lake levels were linked with lower solar activity (Magny, 2004). However, this explanation can only partly be applied to Schweriner See (Fig. 5) and other compared records (Fig. 6). We rather observe temporal offsets between low solar activity and higher lake levels when comparing records from, e.g. Bonk et al. (2023), Pleskot et al. (2018) and Theuerkauf et al. (2022), which might be a result of complex spatial ocean-land interactions as a response to solar activity as suggested by Swindles et al. (2007). For Schweriner See only a few periods align, e.g. the Homeric Grand Solar Minima at ~2800–2550 cal BP (Fig. 5, Fig. 6).

However, more regional to local influences may affect lake-level variability and result in the observed offsets. Instead of solar or climate forcing, Theuerkauf et al. (2022) and Bonk et al. (2023) argue for more local to regional influences on Late Holocene lake levels by identifying (anthropogenic) landcover changes and forest structures as partly responsible for lake-level variations. In particular, a (anthropogenically induced) change between forested and open vegetation landscapes was linked to a changed groundwater recharge and, consequently, higher lake levels under more open vegetation for Tiefer See (~75 ha, ca. 70 km east of Schweriner See, Theuerkauf et al., 2022). Such local-to-regional influences could also have led to varying onsets of lake-level high stands, particularly for smaller lake systems, which are more susceptible to local and regional (anthropogenic) influences. For example, for small Lake Lubińskie (22.7 ha, ca. 275 km southeast of Schweriner See), it is stressed that lake-level variations are mainly related to anthropogenic activity within the catchment (Bonk et al., 2023), which may explain the difference to large Schweriner See. Additional influences, which might lead to different onsets, might be the hydro(geo)logical network or different climatic settings concerning increasing continentality from west to east (Bonk et al., 2023). For Schweriner See, we suggest that such local effects were dampened by the lake's size and minerogenic input reflects mostly lake-level variations related to changes in precipitation and/or evapotranspiration before the 12[th] century. Afterwards, anthropogenic interferences, e.g. weirs, the building of mills or the construction of the Wallenstein trench, influenced the lake level probably beyond natural variations.



## 6 Conclusions

Sediments obtained from Schweriner See are a valuable archive for studying Late Holocene environmental variability. Due to
its size, local (anthropogenic) effects are dampened and proxies reflect large-scale climatic variations, which align well with
interregional paleoclimatic reconstructions covering the Late Holocene. Before 1850 CE, in-lake productivity in Schweriner
See was mainly influenced by winter temperature variability, which modulates ice cover duration and growing season length.
Afterwards, anthropogenic impact on Schweriner See increased significantly, resulting in in-lake productivity mainly driven
by nutrient supply (eutrophication).

Winter temperatures and changes in the moisture source region covary and enable the reconstruction of large-scale atmospheric
processes, suggesting NAO polarity as a driver. Positive NAO conditions are characterized by increased productivity and a
southern moisture source region due to stronger Westerlies bringing warm, moist air towards northwest Europe. In contrast,
conditions resembling a negative NAO (cold and dry) are associated with lower productivity and a northern moisture source
region. Positive NAO conditions occurred from $3030^{+175}/_{-215}$-$2820^{+180}/_{-180}$ cal BP and $2110^{+155}/_{-130}$-$830^{+100}/_{-90}$ cal BP, while
negative NAO-like conditions can be reconstructed for $2820^{+180}/_{-180}$-$2110^{+155}/_{-130}$ cal BP and $830^{+100}/_{-90}$-$105^{+95}/_{-75}$ cal BP. Rates
of changes between positive to negative conditions vary between the individual phases, e.g. with a rapid drop in winter
temperature around $2820^{+180}/_{-180}$ cal BP but a gradual increase from $2110^{+155}/_{-130}$-$1720^{+65}/_{-65}$-cal BP. In addition to these long-
term shifts in atmospheric conditions, short-term hydroclimatic variations can be reconstructed. In this context, titanium mainly
reflects lake-level variations linked to precipitation variability with additional wind speed influences, strengthening wave
action. This mode of minerogenic input contradicts traditional interpretations and highlights the importance of carefully
considering catchment and environmental conditions for proxy interpretation.

### Data availability

The original data from this study will be available in the PANGAEA repository.

### Author Contributions

MLA – Conceptualization, Methodology, Formal analysis, Investigation, Visualization, Writing – original draft preparation,
Writing – review and editing; SC – Investigation, Writing – review and editing; MD – Investigation, Writing – review and
editing; PS – Investigation, Writing – review and editing; MB – Investigation, Writing – review and editing; SL –
Conceptualization, Funding acquisition, Writing – review and editing; MD – Methodology, Resources; TH –
Conceptualization, Methodology, Funding acquisition, Writing – review and editing, Supervision

### Funding

This project was funded by the German Research Foundation (HA5089/14-1) and was carried out in close cooperation with
the Ministry for Climate Protection, Agriculture, Rural Areas and Environment Mecklenburg-Vorpommern.



**Competing Interests**

The contact author has declared that none of the authors has any competing interests

**Acknowledgement**

MLA received a Graduate Scholarship (Landesgraduiertenstipendium) from the Federal State of Mecklenburg-Western Pomerania to conduct this research. We want to acknowledge J. Becker and M. Steinich for their support of the field campaign and M. Steinich and U. Dolgner for CNS and TIC analyses. Moreover, we want to thank Kaja Müller, Antonia Kühn and Rafael Tüllinghoff for their support during sampling and laboratory analyses. We would also like to thank Peter Appleby for carrying out the [137]Cs- and [210]Pb dating and Christian Ohlendorf and Rafael Stiens for XRF scanning. We want to thank Roland
Zech for discussions and laboratory and instrument access at the Physical Geography department of the Friedrich-Schiller University Jena. Moreover, we thank Kevin Jacq for his support during the Hyperspectral Imaging.

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





**Fig. 1: Left inset: Simplified Map of Germany highlighting Schweriner See. A: Digital elevation model of the area surrounding Schweriner See including bathymetry and Weichselian moraines (W1F and W2) surrounding Schweriner See in the north and south. The outlets Wallensteingraben and Stör are indicated in the north and south. The semi-artificial Paulsdamm separates Schweriner See in two similar in size basins, Schweriner Außensee (north) and Schweriner Innensee (south). Although separated, water exchange is still possible (Wöbbecke et al., 2003). Also indicated is the Baltic Sea (B) / North Sea (N) water shed along the eastern and northern shoreline. B: Detailed bathymetric Map of Schweriner See including the coring position (orange star). C: Generalized classification of Schweriner See based on previous investigations on surface sediment samples by Adolph et al. (2023). The eastern, shallow water area is characterized by wave- and wind-induced dynamics (beige). The southern and northern parts are dominated by carbonate precipitation due to increased carbonate-rich groundwater inflow (blue) and productivity (green).**





**Fig. 2: Lithology and sediment colour of the composite record SAS21 (left). A higher organic content causes a darker colour, while a lighter colour is caused by increased carbonate precipitation. The sediment composition is shown as organic matter (=LOI$_{550}$; green), carbonates (= determined by Scheibler method; blue) and residues (yellow). The age-depth model is based on $^{14}$C- (probability density function of the 2σ distribution, blue) and $^{210}$Pb/$^{137}$Cs ages (teal). The mean age and the 95 % confidence interval are shown (centre). $^{210}$Pb/$^{137}$Cs results show a distinct peak for the Chernobyl accident of 1986 (right).**



**Fig. 3: Sedimentological, geochemical, spectral and micropaleontological characteristics of sediment core SAS21. Wave- and wind-induced processes (brown lines) are represented by grain size Mean, Sand$_{63-125\mu m}$, Potassium (K$_{clr}$) and Titanium (Ti$_{clr}$). Iron (Fe$_{clr}$) cannot be assigned to wave- and wind-induced processes or productivity. Productivity (green lines) is shown by Total Organic Carbon (TOC), Total Nitrogen (TN), Loss on ignition 550 °C (LOI$_{550}$), inc/coh ratio, as well as Chlorophyll-a and its derivates (Area$_{600-760}$/R$_{mean}$, 101pt running average). The *n*-alkanes and their isotopic signatures are exemplary (δ²H of *n*C$_{25}$). Carbonate precipitation (blue lines) is represented by the Carbonate content, Total Inorganic Carbon (TIC), Calcium (Ca$_{clr}$), Strontium (Sr$_{clr}$) and the Sr/Ca ratio. Diatom abundance is represented by the percentage of planktonic (teal area) and benthic (light blue area) diatoms, the abundance of eutraphentic diatoms indicating eutrophication and the under-ice blooming diatom *Stephanocostis chantaicus*. Land cover changes are indicated by palynological investigations and represented by the AP/NAP (dark green vs. lime green area) ratio and summed up *Carpinus betulus* and *Fagus sylvatica* (very dark green area). Human impact is represented by ∑(Cu, Ni, Zn) (orange line). XRF data (Ti, K, inc/coh, Ca, Sr, ln(Sr/Ca) and ∑(Cu, Ni, Zn) are shown in 2 mm resolution and as 9pt running average.**







**Fig. 4: Stratigraphic diagram of the past 3070$^{+170}$/$_{210}$ cal BP of SAS21 plotted on an age scale showing sediment colour as an indicator**
**for lithological changes. Ti$_{clr}$ (9pt running average, brown line) indicates paleo-shoreline distance and inc/coh (green line) productivity. Eutraphentic diatoms represent the trophic state based on nutrient supply to Schweriner See, which only increases after 105$^{+95}$/$_{-75}$ cal BP (1845$^{+75}$/$_{-95}$ CE) (Unit F). Diatoms species _Stephanocostis chantaicus_ (teal line) is strictly associated with ice cover duration (Scheffler and Padisák, 2000) and occurs in phases of low productivity. δ²H$_{C25}$ indicates changes in the moisture source region. Land cover is shown by the relation between AP and NAP pollen (dark green vs. light green area). Additionally,**
**changes in the forest composition are represented by the sum of _Carpinus betulus_ and _Fagus sylvatica_ (orange line).**





**Fig. 5: Comparison of hydroclimatic reconstruction from Schweriner See with different archives and solar minima.** Phases of higher/lower lake levels of Schweriner See inferred from (paleo)lacustrine landforms, archaeological findings and historical documents are shown in blue and red (Adolph et al., 2022; Lorenz et al., 2017; Konze, 2017; Umweltministerium Mecklenburg-Vorpommern, 2003) which agree with changes in shoreline distance (brown line, 51 pt average) inferred from $Ti_{clr}$ (this study) and hydroclimatic reconstructions from Dosenmoor (Daley and Barber, 2012) differentiating between drier and wetter conditions. Please note the reversed axis for both parameters. Moisture source region variations modulated by NAO variations are inferred from $\delta^2H_{C25}$, with more negative values suggesting a northwards displacement and/or a lower evaporative enrichment. These variations coincide with variations in winter temperature as inferred from productivity (normalized inc/coh values, green line). The NAO polarity was inferred from distinct changes in $\delta^2H_{C25}$ and inc/coh. Hydroclimatic variations are compared to NAO reconstructions from Norway (Faust et al., 2016), NW-Scotland (Baker et al., 2015) and Central Scandinavia (St. Amour et al., 2010), showing a similar NAO variability over the last 3000 years.



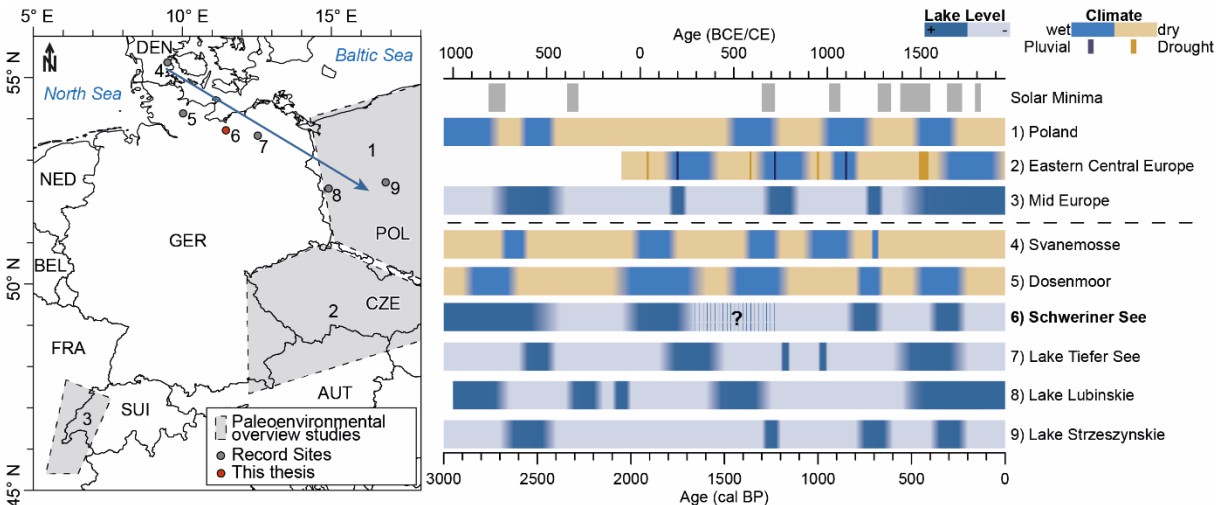

**Fig. 6: Comparison of hydroclimate records covering the past 3000 years. Left: Map of the location of the records. Grey areas indicate the spatial extent of paleoenvironmental overview studies. The blue arrow indicates the NW-SE direction where the compared records are arranged. DEN: Denmark, POL: Poland, CZE: Czechia, AUT: Austria, SUI: Switzerland, FRA: France, BEL: Belgium, NED: Netherlands, GER: Germany. Right: Grouped hydroclimatic records and individual records below the dashed line are shown above. Summarized records are from 1) Poland (Starkel et al., 2013), 2) Eastern Central Europe (Büntgen et al., 2021) and 3) Jura mountains (Magny, 2004). Hydroclimate reconstructions, which show wetter (blue bar) and drier (beige bar) conditions, are compared to lake-level variations and bog surface wetness reflecting hydroclimatic conditions differentiating between lower (light blue bars) and higher (dark blue bars) lake-levels from 4) Svanemosse (Barber et al., 2004), 5) Dosenmoor (Daley and Barber, 2012; Barber et al., 2004), 6) Schweriner See (this study), 7) Tiefer See (Theuerkauf et al., 2022), 8) Lake Lubińskie (Bonk et al., 2023) and 9) Lake Strzeszyńskie (Pleskot et al., 2018). Solar minima are shown as suggested by Usoskin et al. (2007). The question marks and shaded area in the Schweriner See lake-level variations mark the period, masked by increased storminess. The lake level during the period was most likely higher.**