# Peer review of "North Atlantic Oscillation polarity during the past 3000 years derived from sediments of large lowland lake Schweriner See, NE-Germany"

_Climate of the Past, 2023_

## Author Comment (AC1)

**Response to review from Referee 1**

*On behalf of all authors, I would like to thank Referee 1 for the helpful comments and the review of our manuscript. The responses and planned changes are provided in italics and green.*

**General comments:**

The manuscript by Adolph et al. provides a reconstruction of North Atlantic Oscillation (NAO) polarity during the past 3,000 years, derived from lake sediment profiles from Schweriner See, located in NE Germany. The authors employ a multi-proxy approach to interpret past climatic signals. The manuscript presents data of good quality, with clear and organized figures. The supplementary data is well-structured and easy to navigate. In general, the manuscript aligns with the scope of the journal.

However, I have several concerns that should be addressed before publication. Firstly, the authors employ an impressive number of methods (over 10), but many of them lack proper descriptions in the results section, and some are not even mentioned (e.g., grain size analysis). The discussion requires revision since, in its current form, it covers various aspects, such as NAO, climatic events, lake level fluctuations, and minerogenic delivery. However, most of these aspects remain speculative at this point, as the discussion rarely relates to the results. I strongly recommend focusing on the obtained results first, and then comparing them with findings from other sites. The lack of a comprehensive discussion of the results creates the impression that many of the analyses conducted were unnecessary, as their usefulness in the current form of the manuscript is unclear.

*As this was also noted by Reviewer 2, we will restructure the results and discussion section by combining "Results and Interpretation" to better explain the involved processes and improve the Discussion in a revised version of the manuscript.*

**Specific comments:**

The introduction should clearly state the knowledge gap, specific study goals, and hypotheses.

*We agree with the reviewer's suggestion, and we will revise the introduction to adress knowledge gaps, study goals and hypotheses more clearly.*

The term "dominating mode of the NAO" needs a better explanation for clarity.

*We will change the "dominating mode of the NAO" to "respective mode of the NAO"*

Explain the uneven uncertainty of the age-depth model.

*The age-depth model was calculated using the r-package "rbacon". We used the mean and the respective error is based on the 95% confidence interval, in which the probability function is included.*

Rewrite the results section. Currently, it combines results with their interpretation and references to the literature. In this section, only the authors' results should be described. Additionally, the authors have provided over 10 analyses in the methods, some of which are poorly described or not mentioned, such as grain size analysis.

*During the revision, the results section will be rewritten to address this issue and to include all used methods.*

Extending the profile to 1000 cm without time control may not provide a broader perspective. If the extension is relevant, it should be mentioned in the methods section and discussed.

*In a revised version of the manuscript, we will only focus on the upper 9 m.*

Address the discrepancy in the age designation (with changed uncertainty sign) in the introduction part of the discussion.

*To avoid confusion with the changed uncertainty sign and based on the suggestion later in the review, we will remove CE ages.*

Clarify the resolution of distinct analyses and the number of years covered by each sample.
*As shown in Fig. 2, the sedimentation rate changes from 2 to 10 mm a$^{-1}$, which changes also the resolution covered by each analysis. Based on the reviewer's suggestion, we will add the range of years covered by each analysis in a revised version of the paper.*

Discuss whether the diatom signal related to long-lasting ice covers could have been captured for a single, extreme winter event.
*To address this, we will add in the methods section that "For the analysis, one-centimeter-thick samples were used." In the discussion, we will add that "Based on the sample thickness of one centimetre, which covers 1-5 years depending on the sedimentation rate, it is not possible to distinguish between individual years. However, the regularity in the occurrence of S. chantaicus suggests that single events are likely not responsible but rather long-lasting changes in environmental conditions."*

Explain the link between inc/coh and milder winter temperatures, as this ratio was previously associated with lake productivity.
*To account for this suggestion, we will explain the link between inc/coh, productivity and milder winter temperatures in a revised version of the paper in more detail.*

Technical corrections:
Add "years" to the title: "...during the past 3 ka years..."
*Based on the suggestion from reviewer 2, we will change the title to "North Atlantic Oscillation polarity during the past 3000 years derived from sediments of large lowland lake Schweriner See, NE-Germany".*

Use a consistent age unit (CE, cal BP, centuries) for clarity.
*In a revised version, we will use cal BP as a consistent age unit.*

Correct the syntax error in lines 34-37.
*Will be corrected to "Some areas in Central Europe, such as NE-Germany, have already been affected by lowering lake and groundwater levels (Germer et al., 2010)."*

Provide the lengths of cores SAS21-11 and SAS21-12 (line 121).
*We will add the respective lengths of 13.56 and 15.51 m.*

Color is also is one of the sedimentological properties (line 126).
*We will remove the additional mention of sediment colour here.*

Correct the sentence in line 237: "…variations in organic matter variations…".
*Will be changed to "variations in organic matter content"*

Ensure that depth ranges are consistently provided, with the shallower depth mentioned first.
*As the sediment core is described from the bottom to the top, we would like to keep the deeper depth first.*

Line 250 is an interpretation, not a result.
*This will be addressed in a revised version of the manuscript when we revise the results and interpretation section.*

Use consistent language (British English vs. American English) throughout the manuscript.
*BE will be used consistently in a revised version of the manuscript.*

Add a period at the end of the sentence in line 359.
*We will add the period at the end of the sentence.*

Include information on the location of Dosenmoore (line 402).

*We will add "ca. 105 km northwest of Schweriner See" and refer to Fig. 5, where Dosenmoor is shown on a map.*

Add a space between "spread" and the citation (line 438).
*The space will be added.*

It would be good to include map of Europe in Figure 1 for clarity and changing the brackets in the depth scale from () to [] for consistency.
*We will change the Germany inset to a small overview map of Europe as well as the brackets as suggested.*

[Figure]

Ensure consistent terminology in Figure 2 (yellow "remains" vs. "residue").
*This will be changed in a revised version of the manuscript to residues.*

Label the panels in Figure 3 for clarity (e.g., A and B or upper and lower).
*As suggested, A and B will be added to the figure.*

[Figure]

Fig. 1: Sedimentological, geochemical, spectral and micropaleontological characteristics of sediment core SAS21. A) Wave- and wind-induced processes (brown lines) are represented by grain size mean, sand$_{63-125\mu m}$, potassium (K$_{clr}$) and titanium (Ti$_{clr}$). Iron (Fe$_{clr}$) cannot be assigned to wave- and wind-induced processes or productivity. Productivity (green lines) is shown by total organic carbon (TOC), total nitrogen (TN), loss on ignition 550 °C (LOI$_{550}$), inc/coh ratio, as well as chlorophyll-a and its derivates (Area$_{600-760}$/R$_{mean}$, 101pt running average). The n-alkanes and their isotopic signatures are exemplary ($\delta^2H$ of nC$_{25}$). B) Carbonate precipitation (blue lines) is represented by the carbonate content, total inorganic carbon (TIC), calcium (Ca$_{clr}$), strontium (Sr$_{clr}$) and the Sr/Ca ratio. Diatom abundance is represented by the percentage of planktonic (teal area) and benthic (light blue area) diatoms, the abundance of eutraphentic diatoms indicating eutrophication and the under-ice blooming diatom Stephanocostis chantaicus. Land cover changes are indicated by palynological investigations and represented by the AP/NAP (dark green vs. lime green area) ratio and summed up Carpinus betulus and Fagus sylvatica (very dark green area). Human impact is represented by ∑(Cu, Ni, Zn) (orange line). XRF data (Ti, K, inc/coh, Ca, Sr, ln(Sr/Ca) and ∑(Cu, Ni, Zn) are shown in 2 mm resolution and as 9pt running average.

Revise Figure 6 to have consistent numbering for regions (e.g., A, B, and C for Poland, Eastern Central Europe, and Mid Europe). Clarify the difference between "Mid" and "Central."

*We used both terms "Mid Europe" and "Central Europe" as these were the terms used in the original publications. To differentiate and clarify the difference we changed "Mid Europe" to "Jura mountains". Based on the suggestion, we changed the numbering of the regions to A, B and C.*

[Figure]

Fig. 2: Comparison of hydroclimate records covering the past 3000 years. Left: Map of the location of the records. Grey areas indicate the spatial extent of paleoenvironmental overview studies. The blue arrow indicates the NW-SE direction where compared records are arranged. DEN: Denmark, POL: Poland, CZE: Czechia, AUT: Austria, SUI: Switzerland, FRA: France, BEL: Belgium, NED: Netherlands, GER: Germany. Right: Grouped hydroclimatic records and individual records below the dashed line are shown above. Summarized records are from  A) Poland (Starkel et al., 2013), B) Eastern Central Europe (Büntgen et al., 2021) and C) the Jura mountains (Magny, 2004). Hydroclimate reconstructions, which show wetter (blue bar) and drier (beige bar) conditions, are compared to lake-level variations and bog surface wetness reflecting hydroclimatic conditions differentiating between lower (light blue bars) and higher (dark blue bars) lake levels from 1) Svanemosse (Barber et al., 2004), 2) Dosenmoor (Daley and Barber, 2012; Barber et al., 2004), 3) Schweriner See (this study), 4) Tiefer See (Theuerkauf et al., 2022), 5) Lake Lubińskie (Bonk et al., 2023) and 6) Lake Strzeszyńskie (Pleskot et al., 2018). Grand solar minima are shown as suggested by Usoskin et al. (2007). The question marks and shaded area in the Schweriner See lake-level variations mark the period, masked by increased storminess. The lake level during the period was most likely higher.

---

## Author Comment (AC2)

**Response to review from Referee 2**

*On behalf of all authors, I would like to thank Referee 2 for the helpful comments and the review of our manuscript. The responses and planned changes are provided in italics and green.*

The manuscript by Adolph et al. titled 'North Atlantic Oscillation polarity during the past 3 ka derived from lacustrine sediments of large lowland lake Schweriner See, NE Germany' presents a study of a lake sediment core integrating scanning techniques, sedimentological, bulk geochemical, pollen, diatom and leaf wax records. Aim of the study is to reconstruct the environmental factors modifying sediment deposition.

The efforts undertaken are methodologically state of the art and the results provide insights into the regional climate dynamics within the last 3000 years. Therefore, the study can be of interest for a broader geoscience community and would be suitable for publication in Climate of the Past. However, before publication the results/proxy interpretations should be discussed in a more rigorous way, some generalizing statements should be specified or revised and the manuscript would benefit from reorganization.

**Main points:**
I would prefer to read a more focussed, results-based and mechanistic discussion of the possible factors controlling organic matter accumulation, preservation and degradation in Schweriner See and, consequently, the relevance of the area600-700, LOI550, TOC, TN and inc/coh proxies. In this version of the manuscript area600-700 is defined as productivity indicator in the methods section based on one citation (lines 156-157) and LOI550, TOC, TN and inc/coh are defined as productivity proxies based on their correlation with area600-700 in the results section (lines 256-258). Therefore, the presented proxy interpretations and lenghty paleoclimate implications remain ta degree speculative. In addition, the reconstructed NAO polarity and precipitation records from Schweriner See do not match well (e.g. around 700 or 2500 a BP). Please discuss these discrepancies between both proxies, as both should be interconnected. In general, the manuscript would benefit from a clearer distinction between the methods, results and discussion sections.
*As this was also noted by Reviewer 1, we will restructure the results and discussion section by combining "Results and Interpretation" to better explain the involved processes and improve the Discussion in a revised version of the manuscript. We will discuss the discrepancy, though, the addressed "precipitation record" is the minerogenic input, which was not addressed as precipitation but as shoreline distance record. As such it is stated in the manuscript that "the main driver for minerogenic input to the coring location of SAS21 at Schweriner See were lake-level variations with additional wind speed influences and direction amplifying wave action" (Line 520). Phases with additional wind speed influences and wave action were discussed in section 5.2.1 (Processes affecting minerogenic input). We agree with the reviewer that both processes should partly be related. NAO occurs predominantly during winter while lake-level variations may also be significantly influenced by summer droughts, which are not reflected in the NAO proxies. We will address this issue in detail in a revised version of the manuscript.*

**Specific comments:**

Lines 39-40: Continentality is to my knowledge controlled by a place's distance from the ocean and not directly connected with the NAO.
*The reviewer is right and we will rewrite these lines in a revised version of the manuscript to "Recent climate shows a spatial climatic gradient with increasing continentality from west to east and existing paleoenvironmental studies from North Germany point to considerable environmental variability during the Holocene (e.g. Dietze et al., 2016; Theuerkauf et al., 2022; Kaiser et al., 2012)."*

In think the introduction can be streamlined and better organized.
*We will try to reorganize the introduction in a revised version of the manuscript.*

Lines 327-328: This statement is not true. Small lakes do not generally suffer from anthropogenic overprinting. For example, the sediment records from small Lakes Tiefer See, Belau and Woserin located in the Schweriner See region allowed to reconstruct changes in NAO polarity, humidity and wind speed.
*The reviewer is right. We will remove this statement.*

Is the construction work for the Paulsdamm AD 1848 visible in the investigated sediment core? This could be a nice time marker.
*Generally, the decade around 1850 marks a distinct shift in the sedimentation from calcareous to organogenic sediment in Schweriner See, which was previously shown in Adolph et al. (2022). This distinct change was linked to an increase in population density leading to increases in sewage and, consequently, productivity. This distinct shift was observed in short sediment cores from three different locations and also in the record presented in this study. This distinct increase in productivity likely masked the signal of the Paulsdamm construction.*

The lake sediment record investigated in Olsen et al. (2012) is located in Greenland which is not mentioned in the list.
*Greenland will be added to the list in a revised version of the manuscript.*

Since ice cover duration is interpreted to play an important role for productivity changes in Lake Schwerin, it would be interesting to read a sentence about varying ice cover durations during the instrumental period.
*Unfortunately, we do not have any data about the ice cover duration during the instrumental period. However, we would likely not see any interplays between ice cover duration and productivity changes because since 1850 CE productivity is not driven by winter temperature changes but by nutrient availability, which masks the temperature signal. This is shown in this manuscript by the distinct increase in eutraphentic diatoms after 1850 CE as well as in Adolph et al. (2022), which links productivity to sewage disposal and population dynamics within the catchment.*

Lake level reconstruction: Please discuss the role of the Stör river draining Schweriner See for the presented lake level reconstruction. Is the river too small to level out lake level changes?
*We will discuss this in a revised version of the manuscript. But generally, yes, Stör river is a relatively small, shallow river with only a slight gradient. Indeed the river was so shallow that until 1830, Stör river was difficult to navigate by boat, which is why we assume that the river is too small/shallow to level out lake-level changes.*

Lines 403-405. Different moisture sources do not influence the amount of precipitation.
*We will rewrite this section as "As expected under a positive NAO influence, a southern moisture source region is linked to wetter conditions. In comparison, a northern moisture source region under negative NAO conditions causes drier conditions (Fig. 5) due to shifts in the westerly pathway."*

Lines 429-431. This sentence connects a positive NAO polarity with a coinciding period of dryness in Europe. This contradicts with the statement in lines 64-65, associating a positive NAO with more humid conditions.
*We will address this in the revised version of the manuscript.*

Please provide a definition on how you distinguish NAO+ and NAO- time slices based on the Schweriner See data.
*In the revised version of the manuscript, the following section will be added: "NAO time slices are defined by distinct changes in productivity, the occurrence or disappearance of the diatom species S. chantaicus and changes in the compound-specific hydrogen isotopes. Phases with*

*low productivity, which co-occur with the occurrence of S. chantaicus and a shift to depleted δ²H$_{C25}$ values, are defined as negative NAO phases. In contrast, phases with high productivity, which co-occur with the disappearance of S. chantaicus and a shift to enriched δ²H$_{C25}$ values, are defined as positive NAO phases."*

**Detailed comments:**

Line 52: Delete 's' in 'circulations'.
*We will delete this in a revised version.*

Lines 244-245: Shortly mention why 897.5 cm core depth is the lower limit.
*In the revised version of the manuscript, we will add that 897.5 cm core depth is the lower limit because this is the depth of the lowermost $^{14}$C age. We refrain from extrapolating the age-depth model.*

Lines 372-375: Does a distance of 120 km substantially change the degree of continentality and evaporative enrichment?
*Northern Germany has a distinct climatic gradient, which can be observed, i.e., in the water balance. Areas west of Schweriner See have a positive water balance, while areas east of Schweriner See, where the mentioned study was conducted, have a negative water balance (Figure 1) indicating that potential evapotranspiration is higher than precipitation. Therefore, we expect that even these 120 km may have an impact on the degree of continentality and, consequently, the evaporative enrichment.*

[Figure]

*Figure 1: Mean Annual Water Balance (MAWB) of North Germany for 1971-2000 in mm. Areas in red have a negative water balance, and areas in blue have a positive water balance. – Data source: DWD*

Lines 376-386: This part can be shortened, as a detailed description of the NAO is already given in the introduction.
*As suggested, this part will be shortened in a revised version of the manuscript.*

Title: Change '3 ka' to '3000 years' in the title, as ka is not used within the text. Delete 'lacustrine' as lake is mentioned too.
*As suggested, we will change the title to "North Atlantic Oscillation polarity during the past 3000 years derived from sediments of large lowland lake Schweriner See, NE-Germany"*

Fig.1. Except for the coring location is Fig. 1b already included in Fig. 1a. Add the coring location to Fig 1a and delete Fig. 1b?
*We would like to show the bathymetry again separately to highlight the distinct morphometry. In particular, the widespread shallow water area is essential for the discussion of the shoreline distance.*

Fig. 6. Add 'Grand' to 'Solar Minima'.
*As suggested, Grand will be added to the figure.*

[Figure]

*Fig. 1: Comparison of hydroclimate records covering the past 3000 years. Left: Map of the location of the records. Grey areas indicate the spatial extent of paleoenvironmental overview studies. The blue arrow indicates the NW-SE direction where the compared records are located. DEN: Denmark, POL: Poland, CZE: Czechia, AUT: Austria, SUI: Switzerland, FRA: France, BEL: Belgium, NED: Netherlands, GER: Germany. Right: Grouped hydroclimatic records and individual records below the dashed line are shown above. Summarized records are from A) Poland (Starkel et al., 2013), B) Eastern Central Europe (Büntgen et al., 2021) and C) the Jura mountains (Magny, 2004). Hydroclimate reconstructions, which show wetter (blue bar) and drier (beige bar) conditions, are compared to lake-level variations and bog surface wetness reflecting hydroclimatic conditions differentiating between lower (light blue bars) and higher (dark blue bars) lake levels from 1) Svanemosse (Barber et al., 2004), 2) Dosenmoor (Daley and Barber, 2012; Barber et al., 2004), 3) Schweriner See (this study), 4) Tiefer See (Theuerkauf et al., 2022), 5) Lake Lubińskie (Bonk et al., 2023) and 6) Lake Strzeszyńskie (Pleskot et al., 2018). Grand solar minima are shown as suggested by Usoskin et al. (2007). The question marks and shaded area in the Schweriner See lake-level variations mark the period, masked by increased storminess. The lake level during the period was most likely higher.*

---

## Author Response (AR1)

University of Greifswald, Institute of Geography and Geology, Marie-Luise Adolph, 17489 Greifswald

Institute of Geography and Geology

Physical Geography

Dr. Marie-Luise Adolph

Research Associate

Tel: +49 3834 420 4516
marie-luise.adolph@uni-greifswald.de

**Resubmission of revised manuscript cp-2023-73**

Dear Prof. Dr. Piotrowska,
dear Reviewers,

Thank you for the opportunity to revise our paper on "North Atlantic Oscillation polarity during the past 3000 years derived from sediments of large lowland lake Schweriner See, NE-Germany" by Adolph et al. Reviewer suggestions were very helpful, and we also appreciate the reviewers' comments and suggestions.

We completely revised and restructured the manuscript according to your suggestions. The most relevant change made in the manuscript is a restructuring of the Results and Discussion section to Results, Interpretation and Discussion. Please find below a detailed reply to all reviewer comments. The revised versions of the manuscript were uploaded as PDF file with track changes and as PDF file with all changes accepted. Thank you for considering our manuscript for publication.

Sincerely,

Marie-Luise Adolph

**Responses to Reviewer 1**

**General comments:**
The manuscript by Adolph et al. provides a reconstruction of North Atlantic Oscillation (NAO) polarity during the past 3,000 years, derived from lake sediment profiles from Schweriner See, located in NE Germany. The authors employ a multi-proxy approach to interpret past climatic signals. The manuscript presents data of good quality, with clear and organized figures. The supplementary data is well-structured and easy to navigate. In general, the manuscript aligns with the scope of the journal.

However, I have several concerns that should be addressed before publication. Firstly, the authors employ an impressive number of methods (over 10), but many of them lack proper descriptions in the results section, and some are not even mentioned (e.g., grain size analysis). The discussion requires revision since, in its current form, it covers various aspects, such as NAO, climatic events, lake level fluctuations, and minerogenic delivery. However, most of these aspects remain speculative at this point, as the discussion rarely relates to the results. I strongly recommend focusing on the obtained results first, and then comparing them with findings from other sites. The lack of a comprehensive discussion of the results creates the impression that many of the analyses conducted were unnecessary, as their usefulness in the current form of the manuscript is unclear.

*We restructured the Results and Discussion sections into Results (Lines 229-270), Interpretation (Lines 271-430) and Discussion (Lines 431-574) to make the manuscript more consistent and easier to follow for the reader. Moreover, we restructured the Results section to better address the employed methods.*

**Specific comments:**
The introduction should clearly state the knowledge gap, specific study goals, and hypotheses.
*We revised the introduction, restructured it to better address knowledge gaps, study goals and hypothesis.*

The term "dominating mode of the NAO" needs a better explanation for clarity.
*We replaced the term "dominating mode of the NAO" by "respective mode of the NAO" (L. 106)*

Explain the uneven uncertainty of the age-depth model.
*The age-depth model was calculated using the r-package "rbacon". We used the mean and the respective error is based on the 95% confidence interval, in which the probability function is included. We added (Lines 149-141): "In the following, ages are reported as 'rbacon'-derived mean ages and the uncertainty is based on the upper and lower limits of the 95 % confidence interval (Fig. 2)."*

Rewrite the results section. Currently, it combines results with their interpretation and references to the literature. In this section, only the authors' results should be described. Additionally, the authors have provided over 10 analyses in the methods, some of which are poorly described or not mentioned, such as grain size analysis.
*The entire results section, in particular Lines 248-270, were rewritten to better account for all applied methods. We do not examine individual units anymore but rather focus on each parameter to consider all of them adequately.*

Extending the profile to 1000 cm without time control may not provide a broader perspective. If the extension is relevant, it should be mentioned in the methods section and discussed.
*We now only show the upper 9 m, for which we have a robust age control.*

Address the discrepancy in the age designation (with changed uncertainty sign) in the introduction part of the discussion.
*To avoid confusion with the changed uncertainty sign and based on the suggestion later in the review, we removed most CE ages.*

Clarify the resolution of distinct analyses and the number of years covered by each sample.

*As shown in Fig. 2, the sedimentation rate changes from 2 to 10 mm a$^{-1}$, which changes also the resolution covered by each analysis. Based on the suggestion, we added the range of years covered by each analysis in the Methods section, e.g. in Line 164: "Discrete samples were taken in a 1 cm resolution (equivalent to a 1-6 year temporal resolution) using LL-channels (Nakagawa, 2014)."*

Discuss whether the diatom signal related to long-lasting ice covers could have been captured for a single, extreme winter event.

*To address this issue, we added in the methods section that "For diatom analysis, 91 samples with a one-centimetre thickness and 1-2 cm$^3$ volume were taken in the same sampling resolution (equivalent to a 16-85 year temporal resolution) as the pollen analyses." (Lines 212-213). We further added in the interpretation section "Based on the sample thickness for diatom analysis of one centimetre, which covers 1-6 years depending on the sedimentation rate, it is not possible to distinguish between individual years. However, the regularity in the occurrence of S. chantaicus suggests that single events are likely not responsible but rather long-lasting changes in environmental conditions, which is also supported by long-lasting phases of lower productivity during which S. chantaicus occurs (units C and E, Fig. 4)." (Line 304-308)*

Explain the link between inc/coh and milder winter temperatures, as this ratio was previously associated with lake productivity.

*In the revised version, we added section "5.1.1 Organic matter as an indicator for winter temperature variability", in which we explain the link between inc/coh, which is indicative for the organic matter content, productivity and milder winter temperatures in detail, e.g. in Lines 299-304: "Such long-lasting ice-covers under colder winter conditions may substantially affect the seasonal heat budget, timing and length of stratification but also the productivity of aquatic ecosystems (e.g. Bonsal et al., 2006) because long-lasting ice covers delay the onset of the growing season and/or reduce water temperatures, which results in a reduced productivity of the lake system. In contrast, during milder winter temperatures the growing season may start earlier and surface water temperatures may already be increased, which prolongs the growing season and results in a higher productivity of the lake system."*

**Technical corrections:**

Add "years" to the title: "...during the past 3 ka years..."

*Based on the suggestion from reviewer 2, we changed the title to "North Atlantic Oscillation polarity during the past 3000 years derived from sediments of large lowland lake Schweriner See, NE-Germany".*

Use a consistent age unit (CE, cal BP, centuries) for clarity.

*In the revised version, we use cal BP as consistent age unit.*

Correct the syntax error in lines 34-37.

*We corrected the text to "Some areas in Central Europe, such as NE-Germany, have already been affected by lowering lake and groundwater levels (Germer et al., 2010)."*

Provide the lengths of cores SAS21-11 and SAS21-12 (line 121).

*We added the respective lengths of 13.56 and 15.51 m (Line 117).*

Color is also is one of the sedimentological properties (line 126).

*We removed sediment colour here.*

Correct the sentence in line 237: "…variations in organic matter variations…".

*We changed to "variations in organic matter content" (Line 232)*

Ensure that depth ranges are consistently provided, with the shallower depth mentioned first.

*During the revision of the manuscript, we rewrote this section and do not mention specific depths anymore.*

Line 250 is an interpretation, not a result.
*This was addressed in the restructuring of the results section.*

Use consistent language (British English vs. American English) throughout the manuscript.
*BE is used consistently in the revised version of the manuscript.*

Add a period at the end of the sentence in line 359.
*We added the period at the end of the sentence.*

Include information on the location of Dosenmoore (line 402).
*We added "ca. 105 km northwest of Schweriner See" and a reference to Fig. 5, where the location of Dosenmoor is shown on a map. (Line 483)*

Add a space between "spread" and the citation (line 438).
*The space was be added.*

It would be good to include map of Europe in Figure 1 for clarity and changing the brackets in the depth scale from () to [] for consistency.
*Instead of only adding a map of Europe, we included a conceptual overview of the NAO over Europe and added the location of Schweriner See within (Fig. 1A). Additionally, we changed the brackets as suggested (Fig. 1C).*

Ensure consistent terminology in Figure 2 (yellow "remains" vs. "residue").
*We changed it to "remains" for consistency.*

Label the panels in Figure 3 for clarity (e.g., A and B or upper and lower).
*As suggested, A and B was added.*

Revise Figure 6 to have consistent numbering for regions (e.g., A, B, and C for Poland, Eastern Central Europe, and Mid Europe). Clarify the difference between "Mid" and "Central."
*We used both terms "Mid Europe" and "Central Europe" as these were the terms used in the original publications. To differentiate and clarify the difference we changed "Mid Europe" to "Jura mountains". Based on the suggestion, we changed the numbering of the regions to A, B and C.*

**Responses to Reviewer 2**

The manuscript by Adolph et al. titled 'North Atlantic Oscillation polarity during the past 3 ka derived from lacustrine sediments of large lowland lake Schweriner See, NE Germany' presents a study of a lake sediment core integrating scanning techniques, sedimentological, bulk geochemical, pollen, diatom and leaf wax records. Aim of the study is to reconstruct the environmental factors modifying sediment deposition.

The efforts undertaken are methodologically state of the art and the results provide insights into the regional climate dynamics within the last 3000 years. Therefore, the study can be of interest for a broader geoscience community and would be suitable for publication in Climate of the Past. However, before publication the results/proxy interpretations should be discussed in a more rigorous way, some generalizing statements should be specified or revised and the manuscript would benefit from reorganization.

**Main points:**
I would prefer to read a more focussed, results-based and mechanistic discussion of the possible factors controlling organic matter accumulation, preservation and degradation in Schweriner See and, consequently, the relevance of the area600-700, LOI550, TOC, TN and inc/coh proxies. In this version of the manuscript area600-700 is defined as productivity indicator in the methods section based on one citation (lines 156-157) and LOI550, TOC, TN and inc/coh are defined as productivity proxies based on their correlation with area600-700 in the results section (lines 256-258). Therefore, the presented proxy interpretations and lenghty paleoclimate implications remain ta degree speculative. In addition, the reconstructed NAO polarity and precipitation records from Schweriner See do not match well (e.g. around 700 or 2500 a BP). Please discuss these discrepancies between both proxies, as both should be interconnected. In general, the manuscript would benefit from a clearer distinction between the methods, results and discussion sections.

*We restructured the Results and Discussion sections into Results (Lines 229-270), Interpretation (Lines 271-430) and Discussion (Lines 431-574) to make the manuscript more consistent and easier to follow for the reader and to better explain the involved processes (e.g. organic matter accumulation and preservation).*

*Moreover, we added a section within "6.3 Driving mechanisms for lake-level variations" to discuss, among other things, the discrepancy between NAO and lake-level variability (Lines 545-564). Please keep in mind that the addressed "precipitation record" is the minerogenic input, which was not addressed solely as precipitation but as shoreline distance record within additional influences of wind speed changes, e.g. in Lines 428-430 "In conclusion, the main drivers for minerogenic input to the coring location of SAS21 at Schweriner See are shoreline distance variations with additional wind speed influences amplifying wave action, particularly under NAO+ conditions."*

**Specific comments:**

Lines 39-40: Continentality is to my knowledge controlled by a place's distance from the ocean and not directly connected with the NAO.
*We rewrote these lines to "Recent climate in North Germany has a spatial climatic gradient with increasing continentality from west to east." (Line 57)*

In think the introduction can be streamlined and better organized.
*We reorganized the introduction in the revised version of the manuscript.*

Lines 327-328: This statement is not true. Small lakes do not generally suffer from anthropogenic overprinting. For example, the sediment records from small Lakes Tiefer See, Belau and Woserin located in the Schweriner See region allowed to reconstruct changes in NAO polarity, humidity and wind speed.
*We removed this statement.*

Is the construction work for the Paulsdamm AD 1848 visible in the investigated sediment core? This could be a nice time marker.

*Generally, the decade around 1850 marks a distinct shift in the sedimentation from calcareous to organogenic sediment in Schweriner See, which was previously shown in Adolph et al. (2022). This distinct change was linked to an increase in population density leading to increases in sewage and, consequently, productivity. This distinct shift was observed in short sediment cores from three different locations and also in the record presented in this study. This is repeatedly noted in the manuscript e.g. "After 105$^{+95}$/$_{-75}$ cal BP, the anthropogenic impact on Schweriner See increased significantly, resulting in in-lake productivity mainly driven by nutrient supply (eutrophication) masking the hydroclimatic signal." (Lines 589-591). This distinct increase in productivity likely masked the signal of the Paulsdamm construction.*

The lake sediment record investigated in Olsen et al. (2012) is located in Greenland which is not mentioned in the list.

*Greenland was added to the list in the revised version of the manuscript and in Fig. 5.*

Since ice cover duration is interpreted to play an important role for productivity changes in Lake Schwerin, it would be interesting to read a sentence about varying ice cover durations during the instrumental period.

*Unfortunately, we do not have any data about the ice cover duration during the instrumental period. However, we would likely not see any interplays between ice cover duration and productivity changes because since 1850 CE productivity is not driven by winter temperature changes but by nutrient availability, which masks the temperature signal. We added the following sentence: "After 105$^{+95}$/$_{-75}$ cal BP the temperature signal is masked by eutrophication dominating the in-lake productivity (Adolph et al., 2023), which is why it is not possible to link the reconstruction to monitoring data (e.g. ice cover duration)." (Lines 474-476)*

Lake level reconstruction: Please discuss the role of the Stör river draining Schweriner See for the presented lake level reconstruction. Is the river too small to level out lake level changes?

*We added the following part to discuss the Stör river "Previously the second outflow, the Stör waterway, likely had no significant influence on the lake-level because, for example, around 1830 CE, the river was so shallow that it was difficult to navigate the Stör even with boats with shallow drafts (Ruchhöft, 2017). Only the expansion of the Stör waterway in the mid-19$^{th}$ century resulted in a lower lake level afterwards (Fellner, 2007; Umweltministerium Mecklenburg-Vorpommern, 2003), which resulted in the division into the two lake basins we see today (Fig. 1)." (Lines 515-519)*

Lines 403-405. Different moisture sources do not influence the amount of precipitation.

*During the restructuring of the manuscript, we removed those lines.*

Lines 429-431. This sentence connects a positive NAO polarity with a coinciding period of dryness in Europe. This contradicts with the statement in lines 64-65, associating a positive NAO with more humid conditions.

*We rewrote this to "A shift to positive NAO conditions from 2110$^{+160}$/$_{-130}$-830$^{+100}$/$_{-90}$ cal BP with a gradual increase in winter temperature until 1720$^{+70}$/$_{-70}$ cal BP coincides with the Roman Warm Period (RWP, c. 2150-1550 cal BP), which was a period of general warmth in Europe (Lamb, 2013)." (Lines 458-460)*

Please provide a definition on how you distinguish NAO+ and NAO- time slices based on the Schweriner See data.

*We dedicated section "5.1.3 NAO variability" (Lines 348-370) to this issue.*

**Detailed comments:**

Line 52: Delete 's' in 'circulations'.

*We deleted this in the revised version.*

Lines 244-245: Shortly mention why 897.5 cm core depth is the lower limit.

*We added "For this study, only the upper 897.5 cm were investigated in detail as this depth marks the lowermost $^{14}C$ age and we refrained from extrapolating the age-depth model." (Lines 141-143)*

Lines 372-375: Does a distance of 120 km substantially change the degree of continentality and evaporative enrichment?

*During the reorganization, we discussed this issue in detail in lines 553-564: "Such influences of evaporative lake water enrichment have been observed for several smaller lakes in north-eastern Germany (Aichner et al., 2022). However, these study sites are located ca. 120 km southeast of Schweriner See in the more continental climate zone, whereas Schweriner See is located in the transition zone between maritime and continental climate. These areas differ by their mean annual water balance, which is negative in northeast Germany but slightly positive at Schweriner See (Adolph, 2024) suggesting an increased evaporative lake water enrichment in lakes east of Schweriner See. Moreover, lake water evaporation in these lakes shows spatially varying amplitudes and seems to depend on the lake's morphological parameters and hydrological features (Aichner et al., 2022). Additionally, lakes similar to Schweriner See, i.e. deep lakes with high water residence times and absence of river connections, show low evaporative lake water enrichment (Aichner et al., 2022). Because $\delta^2H_{C25}$ mostly correlates to winter temperature changes at Schweriner See (Fig. 5), we suggest that the $\delta^2H_{C25}$ predominantly dependents on moisture source changes in the North Atlantic region potentially explaining differences in the NAO and lake-level reconstructions. Still, an additional influence of evaporative lake water enrichment cannot be completely excluded."*

Lines 376-386: This part can be shortened, as a detailed description of the NAO is already given in the introduction.

*During the revision, we shortened this part.*

Title: Change '3 ka' to '3000 years' in the title, as ka is not used within the text. Delete 'lacustrine' as lake is mentioned too.

*During the revision, we changed the title to "North Atlantic Oscillation polarity during the past 3000 years derived from sediments of large lowland lake Schweriner See, NE-Germany"*

Fig.1. Except for the coring location is Fig. 1b already included in Fig. 1a. Add the coring location to Fig 1a and delete Fig. 1b?

*We would like to show the bathymetry again separately to highlight the distinct morphometry. In particular, the widespread shallow water area is essential for the discussion of the shoreline distance in section 5.2 and 6.2.*

Fig. 6. Add 'Grand' to 'Solar Minima'.

*During the revision, 'Grand' was added to the figure.*

---

## Author Response (AR2)

University of Greifswald, Inst. for Geography and Geology, Marie-Luise Adolph, 17489 Greifswald

Institute for Geography and Geology

Physical Geography

Dr. Marie-Luise Adolph

Research Associate

Tel: +49 3834 420 4516
marie-luise.adolph@uni-greifswald.de

**Resubmission of revised manuscript cp-2023-73**

Dear Prof. Dr. Piotrowska,
dear Reviewers,

Thank you for the opportunity to revise our paper on "North Atlantic Oscillation polarity during the past 3000 years derived from sediments of large lowland lake Schweriner See, NE-Germany" by Adolph et al. again. We appreciate the reviewers' comments and suggestions which were very helpful again.

We completely revised and restructured the manuscript according to your suggestions. The most relevant change made in the manuscript is a restructuring into a Results and a combined Interpretation and Discussion section. Please find below a detailed reply to all reviewer comments. The revised versions of the manuscript were uploaded as PDF file with track changes and as PDF file with all changes accepted. Thank you for considering our manuscript for publication.

Sincerely,

Marie-Luise Adolph

**Responses to Reviewer 1**

**General comments:**
Dear Authors,

Thank you for implementing most of the proposed changes. I can see that the manuscript has improved significantly. However, I still have some minor to major concerns. For example, there is a mix of tenses throughout the manuscript, which is especially evident in the Interpretation chapter. This needs to be corrected.

The entire manuscript should be checked and corrected for language and text flow, as some sentences are challenging to follow (e.g., the sentence in lines 348-355).

While the scientific merit of the manuscript is good, my biggest concern is its extensive length and numerous repetitions. Some information appears first in the introduction, then in the interpretation section, and again in the discussion. I am unsure if this division enhances clarity. It might be beneficial to reorder the manuscript, such as by merging results and interpretation or interpretation and discussion. Additionally, many sentences are overly long, which makes reading difficult.

*During the review we checked the manuscript and corrected for language and text flow, addressed the issue of overly long sentences and removed repetitions. As suggested by Reviewer 2, we restructured the manuscript into a Results (Chapter 4) and Interpretation and Discussion (Chapter 5) section. The overly long sentence mentioned by Reviewer 1 was changed to:*

*"Distinct variations in winter temperatures, moisture source region and/or evaporative lake water enrichment (Fig. 5) are mainly modulated by the North Atlantic Oscillation (NAO) in the North Atlantic region (Hurrell and Deser, 2009). We observe four distinct time slices at Schweriner See: i) From $3030^{+170}/_{-210}$-$2820^{+180}/_{-180}$ cal BP (unit A-B, Fig. 4) and $2110^{+160}/_{-130}$-$830^{+100}/_{-90}$ cal BP (unit D, Fig. 4), milder winter temperatures, a southern moisture source region in the southern/central North Atlantic and/or a higher evaporative lake water enrichment indicate NAO+ conditions. Contrary, ii) from $2820^{+180}/_{-180}$-$2110^{+160}/_{-130}$ cal BP (unit C, Fig. 4) and $830^{+100}/_{-90}$-$105^{+95}/_{-75}$ cal BP (unit E, Fig. 4) colder winter temperatures, a northern moisture source in the northern North Atlantic and/or Arctic regions and/or lower evaporative lake water enrichment correspond to NAO- conditions." (line 340-347)*

It would also be valuable if the authors provided R scripts for age-depth modeling along with the publicly available age-depth model. This would help to verify the model and parameters used, support open data principles, and ensure result repeatability. This is a suggestion only and does not affect the overall rating of the manuscript.

*Thank you so much for this recommendation. We uploaded the rBacon output files in a zip-compressed folder. If something else is needed, please let us know. The $^{14}C$ and $^{210}Pb/^{137}Cs$ results are additionally uploaded in the pdf supplement file.*

Below, I list several **technical issues:**

Line 18: Change "CE" to "calBP."
*Changed to $105^{+95}/_{-75}$ cal BP (~1850 CE)*

Line 22: Rewrite "moisture source region sources" for clarity.
*Changed to "moisture source region"*

Line 23: Clarify the meaning of "i.a."
*Changed to "among others"*

Lines 71 vs 73: Decide between "large lake" and "rather large lake" for consistency.

*Changed to "large lake"*

Line 95: Clarify the missing 7.9% of land cover.
*Added "grassland (7.6 %) and others (0.3 %)"*

Line 175: Define TOC as "Total Organic Carbon (TOC)."
*Added the definition*

Lines 234-235: Move references out of the results section.
*We removed the references in the results section*

Line 244: Ensure consistency in describing depth intervals (44-45 cm instead of 45-44 cm).
*Changed*

Lines 278-280: Divide the long sentence into two for clarity. The second one can start with TOC/N ratio. Additionally, please either add "is" after TOC/N ratio or delete "which" after.
*We divided this sentence into two sentences: "Organic matter parameters agree visually well with in-situ chloropigments ($Area_{600-760}$,* **Fehler! Verweisquelle konnte nicht gefunden werden.***), which are indicative of past primary productivity (van Exem et al., 2022). Additionally, TOC/TN is mostly <12, which suggests a dominance of nonvascular aquatic plants with only a small contribution of vascular plants (Meyers and Ishiwatari, 1993)." (lines 269-272)*

Line 305: Address the discrepancy between the temporal resolutions mentioned. Here, the authors write that the one-centimeter sample in diatom analysis covered 1-6 years, but in the methods section the temporal resolution is equivalent to a 16-85 years. Something is wrong here.
*We addressed this issue in the methods section: "For diatom analysis, 91 samples with a one-centimetre thickness and 1-2 $cm^3$ volume were taken in the same sampling resolution (equivalent to a 16-85 year temporal sampling resolution between samples, with 1 cm samples covering 1-6 years) as the pollen analyses."*

Lines 314-315: Rewrite the sentence for clarity and accuracy. I suggest: Lacustrine sediments generally contain a mixed signal from terrestrial and aquatic sources, which can be distinguished by chain-length distribution of n-alkanes.
*We rewrote the sentence to "Lacustrine sediments generally contain a mixed signal from terrestrial and aquatic sources, which can be distinguished by the n-alkanes chain-length distribution (e.g. Strobel et al., 2021; Ficken et al., 2000)."*

Lines 355-364: Remove the repetition of information from the introduction.
*We removed many information in this section and moved them to the introduction.*

Line 375: Add the missing reference.
*Added*

Thank you for considering these suggestions.

**Responses to Reviewer 2**

**General comments:**
The revised manuscript by Adolph et al. titled 'North Atlantic Oscillation polarity during the past 3000 years derived from sediments of large lowland lake Schweriner See, NE Germany' developed a lot and I support publication of the work in Climate of the Past. Still, the manuscript would benefit from some reorganizations and few improvements.

**Main point:**
The sub-division of the commonly used discussion section in an interpretation (section 5) and discussion (section 6) part leads to repetitions and the reader needs to skip back and forth through the manuscript. Proxy interpretations from the archive itself and comparisons with other paleoclimate archives interact and support each other. Therefore, I suggest to merge sections 5 and 6 and have only one discussion chapter about organic matter/δ2H variations and NAO variability and one discussion chapter about minerogenic input and lake level fluctuations.

*We addressed this issue by combining the Interpretation and Discussion section (Chapter 5), which is discussing NAO variability first and afterwards the lake-level variability.*

**Specific comments:**
(1) Study Area. This chapter can be streamlined, check for repetitions.
*We revised the study area chapter and removed repetitions*

(2) Lines 153-154 (Methods): The interpretation of Area600-700 as proxy for in-situ productivity is presented in the interpretation chapter. Delete it in the methods.
*Deleted*

(3) Line 161-162 (Methods): This interpretation of Cu, Ni, Zn as anthropogenic impact proxy XRF data presented in the interpretation chapter. Please delete it here.
*Deleted*

(4) Lines 170-172. Please quantify the amount of diatoms adding uncertainty to the estimation of siliciclastic matter content. Is this bias relevant for your interpretations?
*We only investigated this qualitatively. We added this information in the methods section. "Subtracting carbonate content and LOI550 from the total sample weight, the percentage of siliciclastics, which includes a share of silicious algae as revealed by qualitative microscopic analyses on the LOI ash residues, was calculated." (Lines 160-161)*

(5) Results chapter 4.2. (Sediment composition): Check for missing references to figures.
*We added missing references*

**Detailed comments:**
Abstract:
Line 23: Replace 'eventually' with 'thereby'?
*We replace it with "thereby"*

Line 26: Use 1850 CE, like in line 18, to be consistent?
*Changed to "105 $^{+95}/_{-75}$ cal BP (~1850 CE)"*

Main text:
Line 48: Add a point after 'conditions' and start a new sentence for NAO-.
Line 48: Add 'more' before 'meridional.
*Added*

Line 122: Delete 'and' before 'photographed'.
*Deleted*

Line 145: 'Clean' instead of 'cling'?
*As we, indeed, used cling wrap, we want to refrain from changing it to "clean wrap".*

Line 150-152: Rephrase: 'Measurements were performed on U-channels…'
*Changed to "Hyperspectral imaging was carried out at the Université Rouen Normandie on U-channels previously extracted from the cores in Greifswald."*

Line 241: Replace 'gravity' by 'short', to be consistent with the methods.
*Replaced*

Line 254: Add 'respectively' after the numbers.
Line 341-343: Add a reference for these sentences.
*Added*

---

## Author Response (AR3)

UNIVERSITÄT GREIFSWALD
Wissen lockt. Seit 1456

University of Greifswald, Inst. for Geography and Geology, Marie-Luise Adolph, 17489 Greifswald

Institute for Geography and Geology

Physical Geography

Dr. Marie-Luise Adolph

Research Associate

Tel: +49 3834 420 4516
marie-luise.adolph@uni-greifswald.de

**Finalized version of revised manuscript cp-2023-73**

Dear Prof. Dr. Piotrowska,
dear Reviewers,

Thank you for accepting our paper on "North Atlantic Oscillation polarity during the past 3000 years derived from sediments of large lowland lake Schweriner See, NE-Germany" by Adolph et al. in Climate of the Past. We added minor corrections according to the reviewers' suggestions to the final version of this manuscript. Please find detailed responses to the reviewer's suggestions below.

Sincerely,

Dr. Marie-Luise Adolph

**Responses to Reviewer 1**

Dear Authors,

Congratulations on revising your manuscript! The manuscript entitled "North Atlantic Oscillation Polarity During the Past 3000 Years Derived from Sediments of Large Lowland Lake Schweriner See, NE-Germany" has significantly improved since the last round of review. It is now a comprehensive and relevant contribution to the topic of hydrological changes. This version is well-structured and reads smoothly.

There are some final mostly technical suggestions:

Please provide depths in order from shallower to deeper. For example, in line 224, it should be 844.5-878.5, not the other way around.

"When a geologist interprets the Earth's history, the story is read from bottom to top." Geological structures are usually described from bottom to top, i.e., from the older to the younger lithological units. Therefore, we would like to keep this traditional concept in geology.

When providing ranges, such as in line 240 ("Grain size means range from 11.56-56.98"), the word "to" should be included. It should read "from 11.56 to 56.98." The same applies to "between," as in line 247 ("ranging between 6.2-21.6"); it should be "between 6.2 and 21.6."

Done

Lines 353-355 largely repeat the first sentence of this paragraph. Please consider rewriting these lines for clarity and conciseness.

Done

**Responses to Reviewer 2**

The structure of the revised manuscript by Adolph et al. titled 'North Atlantic Oscillation polarity during the past 3000 years derived from sediments of large lowland lake Schweriner See, NE Germany' improved significantly and I support publication of the work in Climate of the Past. Some minor modifications should be included before the final publication (see the list below).

Minor comments:

!Line numbers are from the author's tracked changes version!

Abstract:

Line 23: Add 'wind' between 'Westerly' and 'strength'.

Done

Main text:

Line 34: Replace 'i.e.' with 'e.g.'.

Done

Line 45: Delete 'sea-level' and the associated abbreviation. Azores High and Iceland Low are not just sea-level phenomena.

Done

Line 53: Add 'pressure' between 'weaker' and 'gradient'.

Done

Line 54: Add 'and dryer' between 'colder' and 'air'. Delete the following sentence.

Done

Lines 82-84: (from the manuscript)…we hypothesize that the lake is less susceptible to anthropogenic biases that may be experienced when investigating small lacustrine systems and sediments from Schweriner See reflects (supra)regional hydroclimatic variations.

Anthropogenic influences on Ti variability since the 12th century and a human induced eutrophication since AD 1850 that overprints the climate signal are considerable human impacts on sedimentation in Schweriner See. Please modify or delete the above sentence.

This sentence is a hypothesis which is revisited and checked in the conclusions. Therefore, we would like to keep the hypothesis as it is.

Lines 112-113: Delete the sentence about the studied sediment core in the study area chapter. This information is provided a bit later in Chapter 3.1 (Coring and composite profile).

Done

Lines 198-199: Reading that sentence I understand that the pooled 2 cm thick samples result in a 100 to 150-year resolution of the leaf wax time-series. I think you forgot to mention the gaps between the samples.

We revised this section entirely: "For this, one-centimetre-thick samples were taken and pooled with 0.5 cm of sediment above and below the sampling depth. Sample depths were based on significant changes observed in the XRF-scanning results. This resulted in a **sampling distance of 6 to 63.5 cm (equivalent to a 29-195 year temporal sampling resolution, with samples covering 5-10 years)**." (Lines 167-170)

Lines 218-219: Same as above. You do not mention the gaps between the samples (step-size).

We added the step-size.

Lines 231-234: Same as above. Please add a number for the step-size.

We added the step-size.

Line 244: 'boundaries' instead of 'boundary': (Unit C1 to C2 as well as D1, D2 and D3)

Done

Lines 249-257: Provide % values for carbonate and organic matter contents in this result section, when applicable.

We added percentage value when applicable.

Line 258: Add 'radiocarbon' between 'bottommost' and 'sample'.

Done

Line 266: Add a cm value specifying 'Above'.

Done

Line 273: 'sand' not 'Sand'.

We use $Sand_{>125\mu m}$ as an individual proxy, which is also shown in e.g. Fig. 3 and 4. This parameter is also named "$Sand_{>125\mu m}$" in the discussion section. For consistency, we refrain from changing it.

Lines 375-379: Maybe replace the 'hyphens' between the dates with 'to' to improve the readability?

Done

Lines 705-739: The same applies for the time-spans in the conclusions.

Done